# Prompt-augmented Temporal Point Process for Streaming Event Sequence

**Siqiao Xue**[*], **Yan Wang**[*], **Zhixuan Chu**[♠], **Xiaoming Shi, Caigao Jiang, Hongyan Hao**
**Gangwei Jiang, Xiaoyun Feng, James Y. Zhang, Jun Zhou**
Ant Group
Hangzhou, China
{siqiao.xsq,luli.wy,chuzhixuan.czx}@alibaba-inc.com

## Abstract

Neural Temporal Point Processes (TPPs) are the prevalent paradigm for modeling continuous-time event sequences, such as user activities on the web and financial transactions. In real-world applications, event data is typically received in a *streaming* manner, where the distribution of patterns may shift over time. Additionally, *privacy and memory constraints* are commonly observed in practical scenarios, further compounding the challenges. Therefore, the continuous monitoring of a TPP to learn the streaming event sequence is an important yet under-explored problem. Our work paper addresses this challenge by adopting Continual Learning (CL), which makes the model capable of continuously learning a sequence of tasks without catastrophic forgetting under realistic constraints. Correspondingly, we propose a simple yet effective framework, PromptTPP[1], by integrating the base TPP with a continuous-time retrieval prompt pool. The prompts, small learnable parameters, are stored in a memory space and jointly optimized with the base TPP, ensuring that the model learns event streams sequentially without buffering past examples or task-specific attributes. We present a novel and realistic experimental setup for modeling event streams, where PromptTPP consistently achieves state-of-the-art performance across three real user behavior datasets.

## 1 Introduction

Event sequences are ubiquitous in a wide range of applications, such as healthcare, finance, social media, and so on. Neural TPPs (Mei & Eisner, 2017; Shchur et al., 2020; Zuo et al., 2020; Zhang et al., 2020; Yang et al., 2022) have emerged as the dominant paradigm for modeling such data, thanks to their ability to leverage the rich representation power of neural networks. However, most existing works assume a *static* setting, where the TPP model is trained on the entire data, and parameters remain fixed after training. In contrast, real-world event data usually arrives in a *streaming* manner, rendering it impractical to store all data and retrain the model from scratch at each time step due to computational and storage costs. Shown in Figure 1, a common approach is to use sliding windows to frame the data for model training and prediction. Traditional schemes include pretraining a TPP, which is used for all the following test periods, retraining TPP on the data of each slide of windows and online TPPs. However, they either may fail to adapt to new data or suffer from *catastrophic forgetting* (see Appendix A for an empirical analysis).

In our work, we approach the problem by adopting Continual Learning (CL) (Hadsell et al., 2020; Hao et al., 2023; Chu & Li, 2023; Chu et al., 2023b), a relevant area studying how systems learn sequentially from a continuous stream of correlated data. Yet, classical CL models are not fully

---

[*]These authors contributed equally to this work.

[♠]Corresponding author.

[1]Our code is available at https://github.com/yanyanSann/PromptTPP

37th Conference on Neural Information Processing Systems (NeurIPS 2023).

applicable to our problem. A major line of CL methods (Cha et al., 2021; Buzzega et al., 2020) rely on a rehearsal buffer to retrain a portion of past examples. However, they become ineffective when a rehearsal buffer is not allowed – for example, in real-world scenarios where data privacy matters (Shokri & Shmatikov, 2015) or there are resource constraints. Another branch of works (Ke et al., 2020) bypass the forgetting issue by assuming known task identity at test time, but knowing task identity at test time restricts practical usage. Furthermore, the problem of sequential tasks of event sequence in continuous time have barely been studied.

To develop a CL algorithm for such data in real-world scenarios with applicability and generality, we draw inspiration from recent advances in prompt-augmented learning (Liu et al., 2022a; Varshney et al., 2022; Cho et al., 2022; Li et al., 2023; Chu et al., 2023a; Wang et al., 2023). Prompt-augmented learning is a form of machine learning that involves adding additional information or prompts to the training data in order to further improve the performance of the model. This can include adding labels or annotations to the data, providing additional context to help the model better understand the data, or incorporating feedback from human experts to guide the learning process. By incorporating these prompts, the model is able to learn more effectively and make more accurate predictions. Prompt-augmented learning has been used successfully in a variety of applications, including natural language processing, computer vision,

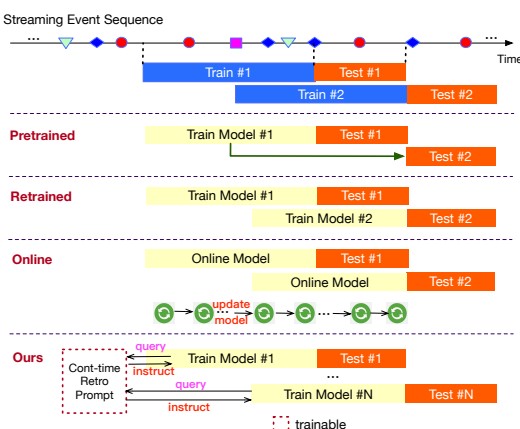

Figure 1: Overview of the classical schemes and PromptTPP framework for streaming event sequences.

and speech recognition. Intuitively, prompt-augmented learning reformulates learning downstream tasks from directly adapting model weights to designing prompts that "instruct" the model to perform tasks conditionally while maintaining model plasticity. Thus, it is promising to leverage prompts to sequentially learn knowledge and further store learned knowledge of event sequence in the CL context. While prompt learning (Wang et al., 2022b,a) already demonstrates its effectiveness on multiple CL benchmarks in language modeling, we wish to extend their success to the models of neural TPPs.

To this end, we propose **PromptTPP**, a novel CL framework whose basis is a **continuous-time retrieval prompt pool** for modeling streaming event sequences. Specifically, we develop a module of *temporal prompt* that learns knowledge and further store the learned knowledge for event sequences in *continuous time*. To improve the applicability, building upon prior works (Wang et al., 2022b), we structure the prompts in a key-value shared memory space called the *retrieval prompt pool*, and design a retrieval mechanism to dynamically lookup a subset of task-relevant prompts based on the instance-wise input of event sequences. The retrieval prompt pool, which is optimized jointly with the generative loss, ensures that shared (unselected) prompts encode shared knowledge for knowledge transfer, and unshared (selected) prompts encode task-specific knowledge that helps maintain model plasticity. PromptTPP has two distinctive characteristics: (i) **applicability**: despite the effectiveness in augmenting TPP with CL, the prompt pool and the event retrieval mechanism removes the necessity of a rehearsal buffer and knowing the task identity, making the method applicable to modeling the event streams in a more realistic CL setting, i.e., memory efficient and task agnostic. (ii) **generality**: our approach is general-purpose in the sense that it can be integrated with any neural TPPs. In summary, our main contributions are:

- We introduce PromptTPP, a novel prompt-augmented CL framework for neural TPPs. It represents a new approach to address the challenges of modeling streaming event sequences by learning a pool of continuous-time retrieval prompts. These prompts serve as parameterized instructions for base TPP models to learn tasks sequentially, thus enhancing the performance of the model.

- We formalize an experimental setup for evaluating the streaming event sequence in the context of CL and demonstrate the effectiveness of our proposed method across three real user datasets.

- By connecting the fields of TPP, CL, and prompt learning, our method provides a different perspective for solving frontier challenges in neural TPPs.

## 2 Preliminaries

**Generative Modeling of Event Sequences.** Suppose we observe $I$ events at a fixed time interval $[0, T]$. Each event is denoted mnemonically as $e@t$ (i.e., "type $e$ at time $t$") and the sequence is denoted as $s_{[0,T]} = [e_1@t_1, \ldots, e_I@t_I]$ where $0 < t_1 < \ldots < t_I \leq T$ and $e_i \in \{1, \ldots, E\}$ is a discrete event type. Note that representations in terms of time $t_i$ and the corresponding inter-event time $\tau_i = t_i - t_{i-1}$ are isomorphic, **we use them interchangeably**.

Generative models of event sequences are TPPs. Specifically, TPPs define functions $\lambda_e$ that determine a finite **intensity** $\lambda_e(t \mid s_{[0,t)}) \geq 0$ for each event type $e$ at each time $t > 0$ such that $p_e(t \mid s_{[0,t)}) = \lambda_e(t \mid s_{[0,t)})dt$. Then the log-likelihood of a TPP given the entire event sequence $s_{[0,T]}$ is

$$\mathcal{L}_{ll} = \sum_{i=1}^{I} \log \lambda_{e_i}(t_i \mid s_{[0,t_i)}) - \int_{t=0}^{T} \sum_{e=1}^{E} \lambda_e(t \mid s_{[0,t)})dt, \tag{1}$$

Instead of posing strong parametric assumptions on the intensity function, neural TPPs (Du et al., 2016; Mei & Eisner, 2017; Zhang et al., 2020; Zuo et al., 2020; Yang et al., 2022) use expressive representations for the intensity function via neural networks and maximize the associated log-likelihood equation 1 via stochastic gradient methods.

**CL Problem Formulation for Streaming Event Sequences.** The typical CL problem is defined as training models on a continuum of data from a sequence of tasks. Given a sequence $s_{[0,T]}$, we split it based on a sliding window approach shown in Figure 1 and form a sequence of tasks over the time $\{\mathcal{D}_0, \ldots, \mathcal{D}_N\}$, where the $\mathcal{T}$-th task $\mathcal{D}_{\mathcal{T}} = (s_{train}^{\mathcal{T}}, s_{test}^{\mathcal{T}})$ contains a tuple of train and test set of event sequences and the two sets have no overlap in time. Data from the previous tasks are not available when training for future tasks. We use the widely-adopted assumption that the task boundaries are clear and the task switch is sudden at training time (Pham et al., 2021). Our goal is to continually learn the sequences while avoiding catastrophic forgetting from the previous tasks.

**Prompt Learning.** Prompt learning methods propose to simply condition frozen language models (LMs) to perform down-stream tasks by learning prompt parameters that are prepended to the input tokens to instruct the model prediction. Compared with ordinary fine-tuning, literature shows In our context, a naive application of prompt learning is to prepend learnable parameters $\boldsymbol{P_s} \in \mathbb{R}^{L_p \times D}$, called a prompt, to the event embedding $\boldsymbol{h} = [\boldsymbol{P_s} || \boldsymbol{x}]$, where $\boldsymbol{x} \in \mathbb{R}^D$ denotes the output of a TPP's embedding layer of an event, and then feed it to the model function $g(\boldsymbol{h})$, i.e., a decoder, to perform downstream tasks. Instead of the native application, in our proposed method, we design a novel prompt learning mechanism to properly model the event streams (see section 3.3).

## 3 Prompt-augmented TPP

We introduce a simple and general prompt-augmented CL framework for neural TPPs, named PromptTPP. As shown in Figure 2, PromptTPP consists of three components: a base TPP model, a pool of continuous-time retrieval prompts and a prompt-event interaction layer. In this section, we omit the task index $\mathcal{T}$ in our notation as our method is general enough to the task-agnostic setting.

### 3.1 Base TPP

A neural TPP model autoregressively generates events one after another via neural networks. For the $i$-th event $e_i@t_i$, it computes the embedding of the event $\boldsymbol{x}_i \in \mathbb{R}^D$ via an embedding layer, which takes the concatenation [2] of the type and temporal embedding $\boldsymbol{x}_i = [\boldsymbol{x}_i^{\text{TYPE}} || \boldsymbol{x}_i^{\text{TEMP}}]$ where $||$ denotes concatenation operation and $\boldsymbol{x}_i^{\text{TYPE}} \in \mathbb{R}^{D_1}, \boldsymbol{x}_i^{\text{TEMP}} \in \mathbb{R}^{D_2}, D = D_1 + D_2$. Then one can draw the next event conditioned on the hidden states that encode history information sequentially:

$$t_{i+1}, e_{i+1} \sim \mathbb{P}_\theta(t_{i+1}, e_{i+1} | \boldsymbol{h}_i), \quad \boldsymbol{h}_i = f_r(\boldsymbol{h}_{i-1}, \boldsymbol{x}_i), \tag{2}$$

where $f_r$ could be either RNN (Du et al., 2016; Mei & Eisner, 2017) or more expressive attention-based recursion layer (Zhang et al., 2020; Zuo et al., 2020; Yang et al., 2022). For the simplicity of notation, we denote the embedding layer and recursion layer together as **the encoder** $f_{\phi_{enc}}$ parameterized by $\phi_{enc}$. Our proposed PromptTPP is general-purpose in the sense that it is straightforward to incorporate any version of neural TPP into the framework.

### 3.2 Continuous-time Retrieval Prompt Pool

The motivations for introducing Continuous-time Retrieval Prompt Pool (**CtRetroPromptPool**) are two-fold. First, existing prompt-learning works focus on classification tasks in NLP or CV domains,

---

[2]The sum operation is also used in some literature. In this paper, we apply concatenation for event embedding.

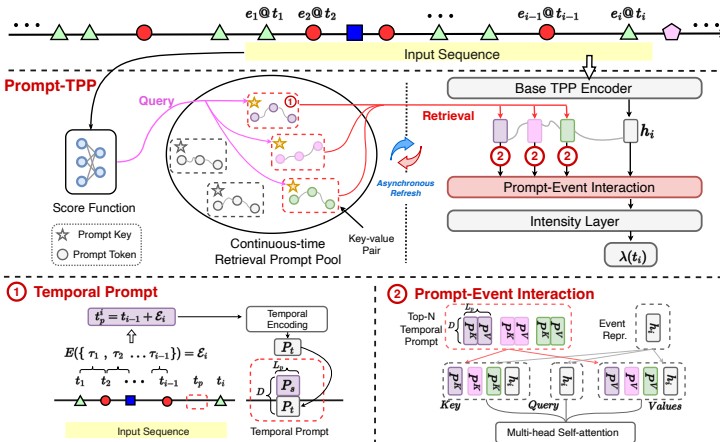

Figure 2: Overview of PromptTPP. Up: At training time, PromptTPP selects a subset of temporal prompts from a key-value paired CtRetroPromptPool based on our proposed retrieval mechanism; then it prepends the selected prompts to the event representations; finally it feeds the extended event representations into the prompt-event interaction and intensity layer, and optimizes the CtRetroPromptPool through the loss defined in equation 11. Down Left: Illustration of how to parameterize a temporal prompt. Down Right: Illustration of prompt tuning in the prompt-event interaction layer.

whose methods are not directly applicable for sequential tasks of learning event streams in continuous time (see section 4.2). Second, the practical setup for modeling event streams closes to the task-agnostic CL setting, where we do not know task identity at test time so that training task-dependent prompt is not feasible. Even if we use extra sources to memorize the task identity, naive usage of prompts (Liu et al., 2022b, 2021; Tam et al., 2022) are still found to result in catastrophic forgetting.

For the first motivation, we construct *temporal prompt* that properly encodes the knowledge of temporal dynamics of event sequence. To address the second, we build a store of prompts in a key-value shared space to transfer knowledge sequentially from one task to another without distinguishing between the common features among tasks versus the features that are unique to each task.

**Temporal Prompt.** In contrast to the standard prompt, the *temporal prompt* is a time-varying learnable matrix that encodes not only the structural but also the temporal knowledge of the event sequence. We define the temporal prompt $\boldsymbol{P} = [\boldsymbol{P}_s; \boldsymbol{P}_t] \in \mathbb{R}^{L_p \times D}$, where $L_p$ is the prompt length and $\boldsymbol{P}_s \in \mathbb{R}^{L_p \times D_1}, \boldsymbol{P}_t \in \mathbb{R}^{L_p \times D_2}$ denotes the structural component and temporal component. While $\boldsymbol{P}_s$ is a learnable submatrix, the temporal component $\boldsymbol{P}_t$ is set to be continuous-time positional encodings of the estimated conditional time so as to consider the timing. More concretely, given $i$-th event, we estimate the arithmetic mean of inter-event times **up to** $t_{i-1}$, denoted by $\mathcal{E}_i = \mathbb{E}[\{\tau_j\}_{j<i}]$ and add this estimated inter-event time to $t_{i-1}$ to get the estimated conditional time $t_p := \widehat{t_i} = t_{i-1} + \mathcal{E}_i$. Inline with Yang et al. (2022), we compute the temporal embedding $\mathrm{TE}(t_p) \in \mathbb{R}^{D_2}$ by

$$\mathrm{TE}(t) = \cos\left(\frac{t}{n_{te}} \cdot \left(\frac{5N_{te}}{n_{te}}\right)^{\frac{d-1}{D_2}}\right) \text{ if } d \text{ is odd}, \mathrm{TE}(t) = \sin\left(\frac{t}{n_{te}} \cdot \left(\frac{5N_{te}}{n_{te}}\right)^{\frac{d}{D_2}}\right) \text{ if } d \text{ is even} \quad (3)$$

where $\{N_{te}, n_{te} \in \mathbb{N}\}$ are hyperparameters selected according to the time scales in different periods. As $\mathrm{TE}(\mathcal{E}_i)$ is a vector, we concatenate it repeatedly to form $\boldsymbol{P}_t$, i.e, $\boldsymbol{P}_t = [\mathrm{TE}(t_p)||,...,||\mathrm{TE}(t_p)] \in \mathbb{R}^{L_p \times D_2}$. Note that the **structural component $\boldsymbol{P}_s$ is learnable while the temporal component $\boldsymbol{P}_t$ is computed deterministically**.

An important consideration of employing such a mechanism is that the mean characterizes the most important property (the long-run average) of the inter-event time distribution, and the computation is straightforward. By taking the temporal embedding of the estimated average conditional time, the prompt efficiently encodes the time-varying knowledge up to the current event, which facilitates learning prediction tasks. We verify the effectiveness of temporal prompt in section 4.2.

**From Prompt to Prompt Pool.** Ideally, one would learn a model that is able to share knowledge when tasks are similar while maintaining knowledge independently otherwise. Thus, instead of

applying a single prompt, we introduce a **pool of temporal prompts** to store encoded knowledge, which can be flexibly grouped as an input to the model. The pool is defined as

$$\boldsymbol{P} = [\boldsymbol{P}_1, ..., \boldsymbol{P}_M], \tag{4}$$

where $M$ denotes the total number of prompts and $\boldsymbol{P}_i \in \mathbb{R}^{L_p \times D}$ is a single temporal prompt. Following the notation in section 3.1, recall $\boldsymbol{h}_i \in \mathbb{R}^D$ denotes the hidden representation of the $i$-th event in the sequence [3] which encodes the event history up to $t_i$ via the recursion by equation 2 and let $\{\boldsymbol{P}_{r_j}, j = 1, ..., N\}$ be a subset of $N$ selected prompts, we then incorporate them into the event sequences as **in-context augmentation** as follows:

$$[\boldsymbol{P}_{r_1}||, ..., ||\boldsymbol{P}_{r_N}||\boldsymbol{h}_i], \tag{5}$$

Prompts are free to compose, so they can jointly encode knowledge for the model to process, which provides flexibility and generality in the sense that a more fine-grained knowledge sharing scheme can be achieved via *prompt retrieval mechanism*. Under this mechanism, a combination of prompts is selected for each task - similar inputs tend to share more common prompts, and vice versa.

**Retrieval Prompt Pool.** The retrieval prompt pool shares some design principles with methods in other fields, such as RETRO (Borgeaud et al., 2022). Specifically, the prompt pool is augmented to be a key-value store $(\mathcal{K}, \mathcal{V})$, defined as the set of learnable keys $\boldsymbol{k} \in \mathbb{R}^D$ and values - temporal prompts $\boldsymbol{P}$ in equation 4:

$$(\mathcal{K}, \mathcal{V}) = \{(\boldsymbol{k}_i, \boldsymbol{P}_i)\}_{i=1}^{M} \tag{6}$$

The retrieval prompt pool may be flexible to edit and can be asynchronously updated during the training procedure. The input sequence itself can decide which prompts to choose through query-key matching. Let $\varphi : \mathbb{R}^D \times \mathbb{R}^D$ be the cosine distance function to score the match between the query and prompt key. Given a query $\boldsymbol{h}_i$, the encoded event vector, we search for the closest keys over $\mathcal{K}$ via maximum inner product search (MIPS). The subset of top-N selected keys is denoted as:

$$\mathrm{K}_{top-N} = \underset{\{r_j\}_{j=1}^{N}}{\operatorname{argmin}} \sum_{i=1}^{N} \varphi(\boldsymbol{h}_i, \boldsymbol{k}_{r_j}) \tag{7}$$

Importantly, the design of this strategy brings two benefits: (i) it decouples the query learning and prompt learning processes, which has been empirically shown to be critical (see section 4.2); (ii) the retrieval is performed in an instance-wise fashion, which makes the framework become *task agnostic*, meaning the method works without needing to store extra information about the task identity at test time. This corresponds to a *realistic setting* for modeling event streams in real applications.

### 3.3 Prompt-Event Interaction

The interaction operation controls the way we combine prompts with the encoded event states, which directly affects how the high-level instructions in prompts interact with low-level representations. Thus, we believe a well-designed prompting function is also vital for the overall CL performance. The interaction mechanism is also called **prompting function** in the NLP community. We apply the multi-head self-attention mechanism (Vaswani et al., 2017) (MHSA) for modeling the interactions and adopt the mainstream realization of prompting function - Prefix Tuning (Pre-T) (Li & Liang, 2021). Denote the input query, key, and values as $\boldsymbol{z}_Q, \boldsymbol{z}_K, \boldsymbol{z}_V$ and the MHSA layer is constructed as:

$$\mathrm{MHSA}(\boldsymbol{z}_Q, \boldsymbol{z}_K, \boldsymbol{z}_V) = [\boldsymbol{z}_1||, ..., ||\boldsymbol{z}_m]\boldsymbol{W}^O, \tag{8}$$

where $\boldsymbol{z}_i = \mathrm{ATTN}(\boldsymbol{z}_Q \boldsymbol{W}_i^Q, \boldsymbol{z}_K \boldsymbol{W}_i^K, \boldsymbol{z}_V \boldsymbol{W}_i^V), \boldsymbol{W}^O, \boldsymbol{W}_i^Q, \boldsymbol{W}_i^K, \boldsymbol{W}_i^V$ are projection matrix. In our context, let $\{\boldsymbol{P}_{r_i}\}_{i=1}^{N}$ be the retrieved prompts from the pool, we set $\boldsymbol{h}_i$ to be the query, split each prompt $\boldsymbol{P}_{r_i}$ into $\boldsymbol{P}_{r_i}^K, \boldsymbol{P}_{r_i}^V \in \mathbb{R}^{L_p/2 \times D}$ and prepend them to keys and values, respectively, while keeping the query as-is:

$$\boldsymbol{h}_i^{Pre-T} = \mathrm{MHSA}(\boldsymbol{h}_i, [\boldsymbol{P}^K||\boldsymbol{h}_i], [\boldsymbol{P}^V||\boldsymbol{h}_i]), \tag{9}$$

where $\boldsymbol{P}^K = [\boldsymbol{P}_{r_1}^K||, ..., ||\boldsymbol{P}_{r_N}^K], \boldsymbol{P}^V = [\boldsymbol{P}_{r_1}^V||, ..., ||\boldsymbol{P}_{r_N}^V]$. Apparently, the key and value $\boldsymbol{z}_K, \boldsymbol{z}_V \in \mathbb{R}^{(\frac{L_p * N}{2} + 1) \times D}$ and the output $\boldsymbol{h}_i^{Pre-T} \in \mathbb{R}^D$. Noted that there exist other prompting methods, such as *Prompt Tuning* (Pro-T), where all the prompts concurrently prepend to the query, key and values:

$$\boldsymbol{h}_i^{Pro-T} = \mathrm{MHSA}([\boldsymbol{P}^Q||\boldsymbol{h}_i], [\boldsymbol{P}^K||\boldsymbol{h}_i], [\boldsymbol{P}^V||\boldsymbol{h}_i]), \tag{10}$$

---

[3]As $D_1 + D_2 = D$, we use $D$ and $(D_1 + D_2)$ interchangeable throughout the paper.

where $\boldsymbol{P}^Q = \boldsymbol{P}^K = \boldsymbol{P}^V = [\boldsymbol{P}_{r_1}||, ..., ||\boldsymbol{P}_{r_N}]$. As a result, the query, key, value and output $\boldsymbol{z}_Q, \boldsymbol{z}_K, \boldsymbol{z}_V, \boldsymbol{h}_i^{Pro-T} \in \mathbb{R}^{(L_p * N+1) \times D}$. Despite being less efficient in computation, we empirically demonstrate that Pre-T brings better performance. See Analysis III in section 4.2.

The output of the MHSA is then passed into an intensity layer (an MLP with softplus activation) to generate the intensity $\lambda_e(t_i), e \in \{1, ..., E\}$. For simplicity, we denote the prompt-event interaction and intensity layer together as the **the decoder** $f_{\phi_{dec}}$ parameterized by $\phi_{dec}$.

## 3.4 Model Optimization

The full picture of PromptTPP at training and test time is described in Algorithm 1 and Algorithm 2 in Appendix C.1. At every training step, each event $e_i@t_i$ is recursively fed into the encoder $f_{\phi_{enc}}$, after selecting $N$ prompts following the aforementioned retrieval strategy, the intensity $\boldsymbol{\lambda}(t_i)$ is computed by the decoder $f_{\phi_{dec}}$. Overall, we seek to minimize the end-to-end loss function:

$$\min_{\boldsymbol{P}, \phi_{enc}, \phi_{dec}, \mathcal{K}} \mathcal{L}_{nll}(\boldsymbol{P}, f_{\phi_{enc}}, f_{\phi_{dec}}) + \alpha \sum_i \sum_{\mathrm{K}_{top-N}} \varphi(f_{\phi_{enc}}(e_i@t_i), \boldsymbol{k}_{r_j}), \tag{11}$$

where the first term is the negative loglikelihood of the event sequence ($\mathcal{L}_{nll}$ equals to $-\mathcal{L}_{ll}$ defined in equation 1) and the second term refers to a surrogate loss to pull selected keys closer to corresponding query in the retrieval process. $\alpha$ is a scalar to control the importance of the surrogate loss. Given the learned parameters, we may wish to make a minimum Bayes risk prediction about the next event via the thinning algorithm (Mei & Eisner, 2017; Yang et al., 2022).

**Asynchronous Refresh of Prompt Pool.** The prompts may lead to the variable contextual representation of the event as the parameters of the based model are continually updated. To accelerate training, we propose to asynchronously update all embeddings in the prompt pool every $C$ training epochs.

# 4 Experiments

## 4.1 Experimental setup

**Datasets and Evaluation Setup** We conduct our real-world experiments on three sequential user-behavior datasets. In each dataset, a sequence is defined as the records pertaining to a single individual. The **Taobao** (Alibaba, 2018) dataset contains time-stamped user click behaviors on Taobao shopping pages with the category of the item involved noted as the event type. The **Amazon** (Ni, 2018) dataset contains time-stamped records of user-generated reviews of clothing, shoes, and jewelry with the category of the reviewed product defined as the event type. The **StackOverflow** (Leskovec & Krevl, 2014) dataset contains two years of user awards on a question-answering website: each user received a sequence of badges with the category of the badges defined as the event type. See Appendix D.1 for dataset details.

We partition Taobao and Amazon datasets into 10 consecutively rolling slides (namely 10 tasks) and partition the StackOverflow dataset into 6 rolling slides (namely 6 tasks). For the Taobao dataset, each slide covers approximately 1 day of time; for the Amazon dataset, each slide covers 2 years of time; for the StackOverflow dataset, each slide covers approximately 5 months time. The subset in each task is split into training, validation, and test sets with a 70%, 10%, 20% ratio by chronological order. Each task has no overlap in the test set. For a detailed discussion, a demonstration of the evaluation process is provided in Figure 9 in Appendix D.3.

**Metrics.** Following the common next-event prediction task in TPPs (Du et al., 2016; Mei & Eisner, 2017), each model attempts to predict every held-out event $(t_i, k_i)$ from its history $\mathcal{H}_i$. We evaluate the prediction $\hat{k}_i$ with the error rate and evaluate the prediction $\hat{t}_i$ with the RMSE.

**Base models.** While our proposed methods are amenable to neural TPPs of arbitrary structure, we choose two strong neural TPPs as our base models: **NHP** (Mei & Eisner, 2017) and **AttNHP** (Yang et al., 2022), an attention-based TPP whose performance is comparable to or better than that of the NHP as well as other attention-based models (Zuo et al., 2020; Zhang et al., 2020).

**Competitors.** With NHP and AttNHP as base models, we trained *PromptNHP* (**Pt-NHP**) and *PromptAttNHP* (**Pt-ANHP**) in the proposed prompt-augmented setup and compared with 7 baselines.

- *PretrainedTPP*. *PretrainedNHP* (**Pre-NHP**) and *PretrainedAttNHP* (**Pre-ANHP**) represent NHP and AttNHP learned at the first task (time step) and not trained any longer.

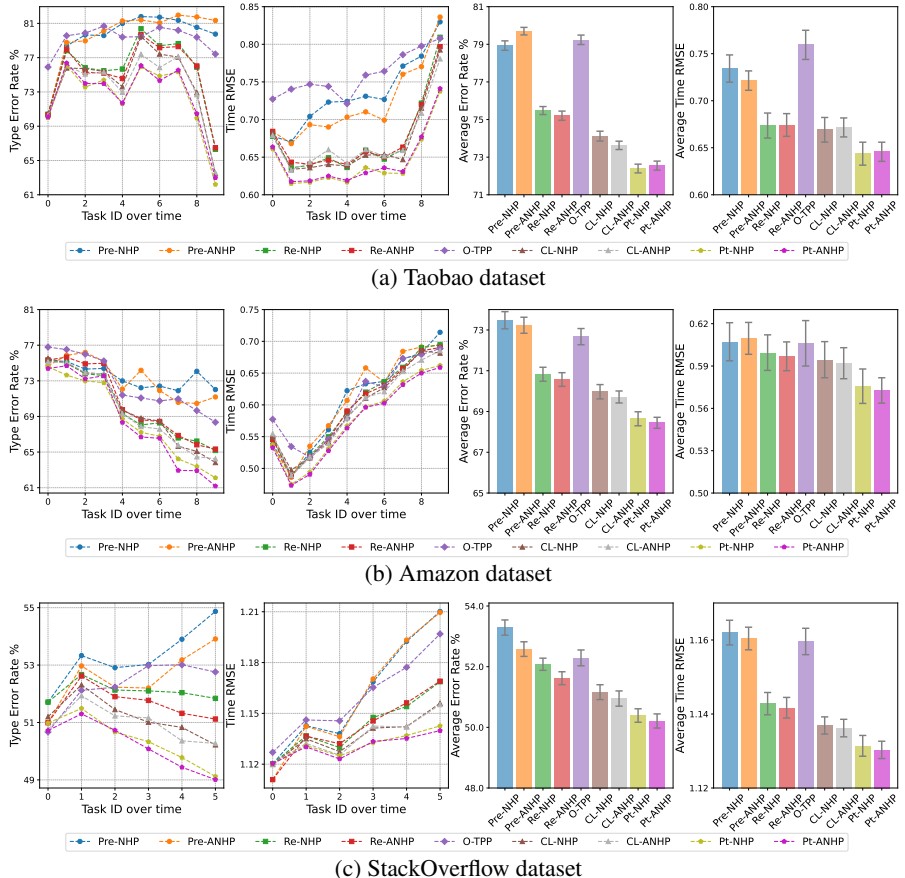

(a) Taobao dataset

(b) Amazon dataset

(c) StackOverflow dataset

Figure 3: Performance of all the methods on Taobao (up), Amazon (middle) and StackOverflow (down). In each figure, the subfigures from left to right are the evolution of type error rate and the time RMSE of each task, the average error rate, and the average time RMSE of all the tasks.

- *RetrainedTPP*. *RetrainedNHP* (**Re-NHP**) and *RetrainedAttNHP* (**Re-ANHP**) refer to TPPs retrained at every sliding widow.

- *OnlineTPP*. As there is no prior work on online neural TPPs, we use online Hawkes process *OnlineMHP* (**O-TPP**) (Yang et al., 2017), trained in an online manner without any consideration for knowledge consolidation.

- *CLTPP*. The concurrent work (Dubey et al., 2022), to the best of our knowledge, is the only neural TPP with CL abilities proposed so far. Based on their work [4], we implement *CL-NHP* (**CL-NHP**) and *CLAttNHP* (**CL-ANHP**) as two variants of the hypernetwork-based CLTPPs.

**Implementation and Training Details.** For a fair comparison, they (except O-TPP which is a classical TPP model) are of similar model size (see Table 2 in Appendix D.4). For Pt-NHP and Pt-ANHN, we set $M = 10, N = 4, L_p = 10$ for both datasets. During training, we set $C = 2$ by default and explore the effect of asynchronous training in Analysis IV of section 4.2. More details of the implementation and training of all the methods are in Appendix D.5.

## 4.2 Results and Analysis

The main results are shown in Figure 3. Pre-NHP and Pre-ANHP work the worst in most cases because of inadequate ability to handle the distribution shift in the event sequence. Besides. O-TPP has a similarly poor performance because of two reasons: first it is a classical (non-neural) TPP with weaker representation power of modeling event sequence compared to its neural counterparts; second as a traditional online learning method, it easily loses memory of previously encountered data and suffers from *catastrophic forgetting*. Retraining at every task (Re-NHP and Re-ANHP) achieves

---
[4]They have not published the code yet.

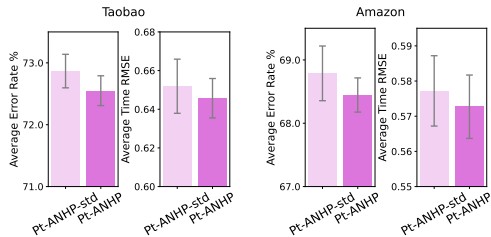

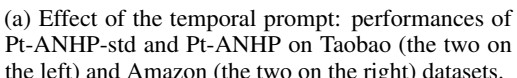

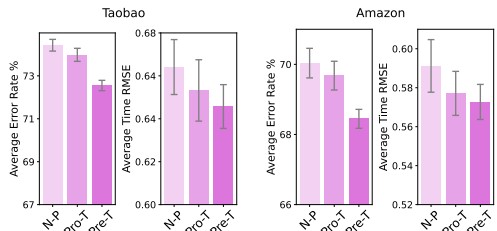

(a) Effect of the temporal prompt: performances of Pt-ANHP-std and Pt-ANHP on Taobao (the two on the left) and Amazon (the two on the right) datasets.

(b) Effect of prompting functions: performances of N-P, Pro-T, and Pre-T on Taobao (the two on the left) and Amazon (the two on the right) datasets.

Figure 5: Effect of temporal prompt and prompting function of PromptTPP.

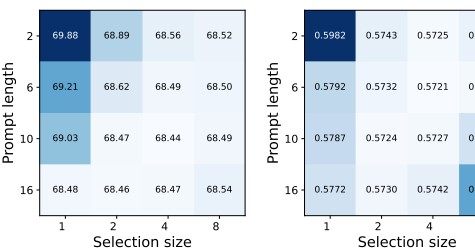

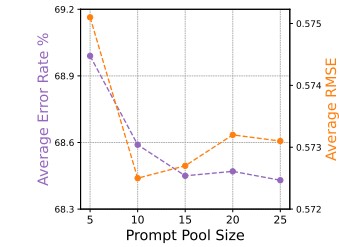

(a) Average performance of Pt-ANHP w.r.t. prompt length $L_p$ and prompt selection size $N$, given prompt pool size $M = 15$. Left: Average Type error rate (%); Right: Average time RMSE.

(b) Average performance of Pt-ANHP w.r.t. prompt pool size $M$, given $L_p = 10$, $N = 4$.

Figure 6: Effect of hyperparameters of PromptTPP on Amazon dataset.

moderate results but it also causes *catastrophic forgetting*. Not surprisingly, CL-NHP and CL-ANHP perform better than retraining, by applying a regularized hypernetwork to avoid forgetting. However, the hypernetwork relies on task descriptors built upon rich meta data, which limits its applicability and performance in our setup (and in real applications as well!). Lastly, our methods (both Pt-NHP and Pt-ANHP) work significantly better than all these baselines across the three datasets: they substantially beat the non-CL methods; they also consistently outperform CL-NHP and CL-ANHP by a relative $4\% - 6\%$ margin on both metrics, thanks to our novel design of the CtRetroPromptPool, which successfully reduces catastrophic forgetting (see Analysis 0).

**Analysis 0: How models perform on previous tasks after learning new events?** We aim to validate that the improvement in performances indeed is due to the alleviation in catastrophic forgetting instead of simply a better fit on the current task. We use ANHP trained on task 9 and Pt-ANHP *continuously* trained on task 9, re-evaluate them on previous tasks and see how the metrics changed. Specifically, on Figure 4,

(i) Each number on the curves of Re-ANHP and CL-ANHP corresponds to the performance difference on the test set of task $i, i < 9$ using ANHP trained on task 9 vs ANHP trained on task $i$.

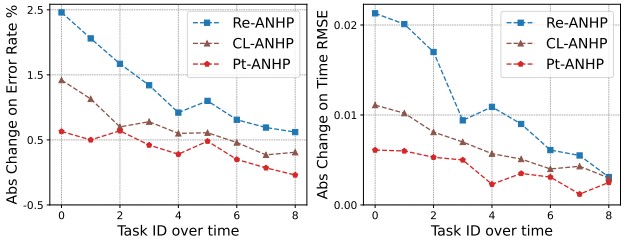

Figure 4: The performance drop when re-evaluating the 0-8-th tasks using model trained on the 9-th task on Amazon dataset.

(ii) Each number on the curves of Pt-ANHP corresponds to the performance difference on the test set of task $i, i < 9$ using Pt-ANHP trained until (including) task 9 vs Pt-ANHP trained until task $i$.

See from Figure 4, on both metrics, we see the drop in performance (i.e., error rate / RMSE increases) of Pt-ANHP is much less significant than ANHP, indicating Pt-ANHP stores well the knowledge of previous tasks, which largely alleviates catastrophic forgetting.

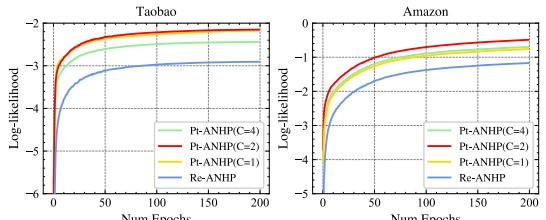
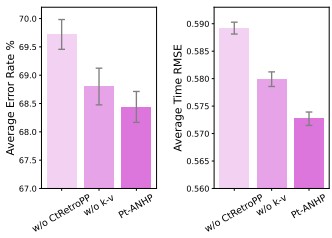

(a) Learning curves of Re-ANHP and Pt-ANHP with varying asynchronous refresh parameter $C$ on a randomly selected task.

(b) Effect of prompt related components of PromptTPP on Amazon dataset.

Figure 7: Effect of asynchronous refresh and prompt related components of PromptTPP on dataset.

**Analysis I: Does stronger base TPP model naively improve CL?** Our method builds upon a backbone TPP and understanding this question is important for fair comparisons and future research. From Figure 3, Re-ANHP makes no consistent improvement against Re-NHP on average CL performance, which indicates a stronger TPP is not a solution for CL without being appropriately leveraged. Besides, for the CL-based methods, CL-ANHP is tied with CL-NHP on Taobao and makes a limited advancement against CL-NHP on Amazon, while Pt-NHP and Pt-ANHP perform closely on both datasets. Therefore, we can conclude that, although AttNHP is a more robust base model than common non attention-based TPP, i.e., NHP, it is not necessarily translated to CL performance.

**Analysis II: Temporal prompt vs standard prompt.** For a fair comparison, we initialize a pool of standard prompts without time-varying parameters by fixing their temporal components $P_t$ to be an all-ones matrix and incorporate it into the base model AttNHP. This method is named Pt-ANHP-std. With other components fixed, we compare Pt-ANHP-std with Pt-ANHP to validate the effectiveness of the temporal prompt introduced in section 3.2.

Figure 5a shows that Pt-ANHP achieves better performance on both datasets: the introduction of temporal prompts slightly improves the RMSE metric and reduces the error rate with a larger margin. We did the paired permutation test to verify the statistical significance of the improvements. See Appendix D.6 for details. Overall, on both datasets, we find that the performance improvements by using the temporal prompts are enormously significant on error rate (p-value $< 0.05$ ) and weakly significant on RMSE (p-value $\approx 0.08$).

**Analysis III: How to better attach prompts?** We explore how to attach prompts and enhance their influences on overall performance. We compare three types of prompting: ① *Naive Prompting* (N-P), where the retrieval and prompting are performed after the event embedding layer: we replace $h_i$ with $x_i$ in equation 7, prepend the selected prompts to $x_i$ and pass it to the rest structure of TPP. ② *Prompt Tuning* (Pro-T): which concurrently prepend the prompts to query, key, and value, introduced at the end of section 3.3. ③ *Prefix-Tuning* (Pre-T), proposed in the main body of section 3.3, which is the **prompting method used in PromptTPP**.

In Figure 5b, we observe that Pre-T leads to better performance on both datasets compared to those two variants. Despite its empirically better performance, the architecture of Pre-T is actually more scalable and efficient when attached to multiple layers since it results in unchanged output size: $h_i^{Pre-T} \in \mathbb{R}^D$ remains the same size as the input while $h_i^{Pro-T} \in \mathbb{R}^{(L_p * N+1) \times D}$ increases the size along the prompt length dimension.

**Analysis IV: Efficiency of our method.** We examine the efficiency of our method in two steps:

- Firstly, seen from Table 2, on both datasets, Pt-NHP / Pt-ANHP leads to a $12\%$ / $8\%$ total parameter increase to the base model, which in fact causes a marginal impact on training speed: Figure 7a shows that learning curves of Re-ANHP and Pt-ANHP($C=1$) converge at almost the same speed to achieve competitive log-likelihood, respectively.

- Furthermore, to accelerate the training (especially when introducing large size prompts), we introduce the asynchronous refresh mechanism (see section 3.4) with prompts updated in a frequency $C > 1$ (refresh the prompt pool less frequently). We observe in Figure 7a that Taobao training with $C = 2$ has a comparable performance with $C = 1$ while Amazon training with $C = 2$ improves the convergence notably. $C = 4$ leads to no advancement.

Overall, PromptTPP only adds a small number of parameters so that it generally has the same convergence rate as the base model. The asynchronous prompt optimization scheme with $C = 2$ improves the convergence more remarkably on the Amazon dataset. In addition, we indeed provide a complexity analysis. See Appendix D.7.

**Analysis V: Effect of prompt related components of our method.** Firstly we completely remove the CtRetroPromptPool design (*w/o CtRroPP* in Figure 7b) and use a single temporal prompt to train tasks sequentially. The performance declines with a notable drop, indicating that a single prompt suffers severe catastrophic forgetting between tasks, while our design of CtRetroPromptPool encodes task-invariant and task-specific knowledge well. Secondly, we remove the learnable key associated with prompts (*w/o k-v* in Figure 7b) and directly use the mean of prompts as keys. This strategy causes a moderate drop in performance. To conclude, learnable keys decouple the query and prompt learning processes and markedly contribute to the performance.

**Analysis VI: Effect of hyperparameters of our method.** We evaluate how the performance of PromptTPP changes as we vary three key hyperparameters: (i) prompt length $L_p$, (ii) selection size $N$, and (iii) prompt pool size $M$. Theoretically, $L_p$ determines the capacity of a single prompt (which jointly encodes certain knowledge), $L_p \times N$ is the total size used to prepend the event vector, while $M$ sets the up limit of the capacity of learnable prompts.

- *Prompt length $L_p$ and selection size $N$.* From the results in Figure 6a, a too small $L_p$ negatively affects results as a single prompt has a too limited ability to encode the knowledge. Besides, given an optimal $L_p$, an overly large $N$ makes the total prompts excessively oversized, leading to underfitting and negatively impacting the results. We conclude that a reasonably large $L_p$ and $N$ enable the model properly encode the shared knowledge between the tasks of event sequences and substantially improve the predictive performance.
- *Prompt pool size $M$.* Figure 6b illustrates that $M$ positively contributes to the performance. This is because the larger pool size means the larger capacity of the prompts.

## 5 Conclusion

In summary, this paper has proposed a groundbreaking framework, known as PromptTPP, for modeling streaming event sequences. By incorporating a continuous-time retrieval prompt pool, the framework effectively facilitates the learning of event streams without requiring rehearsal or task identification. Our experiments have shown that PromptTPP performs exceptionally well compared to other competitors, even under challenging and realistic conditions.

## 6 Limitations and Societal Impacts

**Limitations.** Our method uses neural networks, which are typically data-hungry. Although it worked well in our experiments, it might still suffer compared to non-neural models if starved of data.

**Societal Impacts.** By describing the model and releasing code, we hope to facilitate probabilistic modeling of continuous-time sequential data in many domains. However, our model may be applied to unethical ends. For example, it may be used for unwanted tracking of individual behavior.

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

# Appendices

## A Discussion on Classical Schemes for Modeling Event Sequence

As shown in Figure 1, a common approach is to use sliding windows to frame the data for model training and prediction. In this setup, we discuss three classical schemes.

- *Pretrained TPP.* A straightforward solution is to pretrain a TPP in the first train set and use it for all the following test periods. Such an approach faces the problem of distribution shift, i.e., the new data is systematically different from the data the model was trained on Snoek et al. (2019). Take the Taobao dataset (Alibaba, 2018) for example, which contains time-stamped user click behaviors [5]. We split the dataset by timestamps into 10 periods sequentially and compute the 3S statistics (Shchur et al., 2021) as a measure of the distribution of event sequences for each period. Figure 8a shows that the 3S statistics differentiate between periods while Figure 8b illustrates an increasing KL divergence of the 3S statistics between the first and later period, implying a pattern shift over time. As a result, this approach may fail to adapt to new data and produce unsatisfactory predictions.

- *Retrained TPP.* Another classical solution is to train a new TPP on the data of sliding windows over again. The TPP can quickly adapt to new data but may suffer from *catastrophic forgetting* (McCloskey & Cohen, 1989): adaptation usually implies that the model loses memory of previously encountered data that may be relevant to future predictions. For example, Figure 8a shows a large overlap in distributions of different periods on the Taobao dataset, indicating the necessity of maintaining the knowledge of existing patterns to improve generalization (Snoek et al., 2019; Wang et al., 2020).

- *Online TPP.* A better solution is an online approach: discretize the time axis into small intervals and then incrementally update the TPP at the end of each interval using an online algorithm. However, online models are generally more difficult to maintain and may also cause *catastrophic forgetting* (Hoi et al., 2021). Besides, to the best of our knowledge, apart from online classical TPPs (Yang et al., 2017; Hall & Willett, 2016), the field of online neural TPPs is much less well-studied.

## B Related Work Details

Here we draw connections and discuss differences between our method to related works.

**Temporal Point Process.** A large variety of Neural TPPs have been proposed over recent decades, aimed at modeling event sequences with varying sorts of properties. Many of them are built on recurrent neural networks (Du et al., 2016; Mei & Eisner, 2017; Xiao et al., 2017; Omi et al., 2019; Shchur et al., 2020; Mei et al., 2020; Boyd et al., 2020). Models of this kind enjoy continuous state spaces and flexible transition functions, thus achieving superior performance on many real-world datasets, compared to classical models such as the Hawkes process (Hawkes, 1971). To properly capture the long-range dependency in the sequence, the attention mechanism (Vaswani et al., 2017) has been adapted to TPPs (Zuo et al., 2020; Zhang et al., 2020; Xue et al., 2021; Qu et al., 2023; **?**; Wang et al., 2023; Shi et al., 2023) to enhance the predictive performance. However, learning the event sequence under the *stream* setting is largely unexplored. To the best of our knowledge, there exist two prior works (Yang et al., 2017; Hall & Willett, 2016) that propose online learning algorithms for classical TPPs while that for neural TPP have rarely been studied. We show our method works better than classical online TPPs in practice (see section 4.2).

**Continual Learning.** There is also a rich existing literature on CL: the models can be categorized into *regularization-based* methods (Kirkpatrick et al., 2017; Zenke et al., 2017), which regularize important parameters for learned tasks, *architecture-based* methods (Rusu et al., 2016; Mallya & Lazebnik, 2018) which assign isolated parameters for each task and *rehearsal-based* methods (Cha et al., 2021; Buzzega et al., 2020) which save data from learned tasks in a rehearsal buffer to train with

---

[5]Please see Appendix D.1 for the details of the Taobao dataset and Appendix D.2 for the explanations on 3S statistics and the procedure of the experiment on distribution shift.

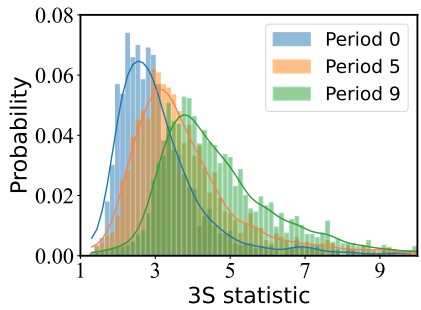
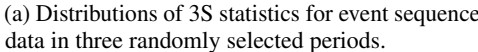
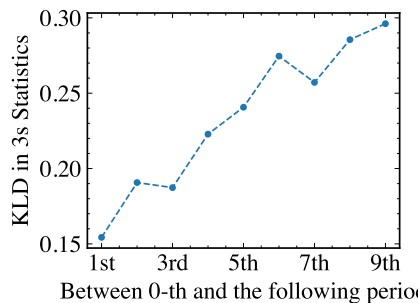

(a) Distributions of 3S statistics for event sequence data in three randomly selected periods.

(b) KL divergence of distributions of 3S statistics between the 0-th and the following period.

Figure 8: Analysis of distribution shift on Taobao dataset.

the current task. In retrospect, we realize a concurrent work (Dubey et al., 2022) which also augments TPP with CL abilities. Important distinctions of their work from ours include: 1. setup: they use standard TPP train/valid setting while we formalize more realistic streaming setting to train/validate the models; 2. methodology: they model the event streams with a hypernetwork-based regularizer while we use a trainable prompt pool with more flexibility and generality in CL. 3. task agnostic: they rely on a task descriptor built from meta attributes of events while we do not - our method is task agnostic. As their source code is not available yet, we independently implement it and our method still outperforms it (see section 4.2).

**Prompt Learning.** Prompt-based learning (or prompting), as an emerging transfer learning technique in NLP, applies a fixed function to condition the model so that the language model gets additional instructions to perform the downstream task. Continuous prompts have also been proposed (Lester et al., 2021; Li & Liang, 2021) to reduce prompt engineering, which directly appends a series of learnable embeddings as prompts into the input sequence, achieving outstanding performance on transfer learning. Wang et al. (2022a,b) connect prompting and CL, which attaches prompts to the pretrained backbone to learn task-invariant and task-specific instructions. Note that it is non-trivial to apply prompting to neural TPPs (See Analysis II and III in section 4.2), and our proposed novel framework reveals its values to event sequence modeling.

## C   Method Details

### C.1   PromptTPP at Training and Test Time

The training and test time Algorithms for PromptTPP are illustrated in Algorithm 1 and Algorithm 2, respectively.

For simplicity of notations, in test time, we show how to sample the next event given one historical event sequence via the thinning algorithm (Mei & Eisner, 2017), which can be easily extended to batch-wise inference (see the implementation in EasyTPP (Xue et al., 2023)).

## D   Experimental Details

### D.1   Dataset Details

We evaluate our methods on three industrial user behavior datasets. We provide details on the preparation and utilization of each below. For both datasets, users are associated with anonymous aliases to remove personally identifiable information (PII).

**Taobao** (Alibaba, 2018). This dataset contains time-stamped user click behaviors on Taobao shopping pages from November 25 to December 03, 2017. Each user has a sequence of item click events where each event contains the timestamp and the category of the item. Following the previous work (Xue et al., 2022), the categories of all items are first ranked by frequencies, and the top 19 are kept while the rest are merged into one category, with each category corresponding to an event type. We work

---
**Algorithm 1** PromptTPP at training time of the $\mathcal{T}$-th task.
---
**Input:** Train set $\{s_{train}\}$. CtRetroPromptPool $(\mathcal{K}, \mathcal{V}) = \{(\boldsymbol{k}_i, \boldsymbol{P}_i)\}_{i=1}^M$ (inherited from the previous task) , score function $\varphi$, loss weight $\alpha$ and asynchronous refresh frequency $C$.

**Output:** Trained base model with a encoder $f_{\phi_{enc}}$ and a decoder $f_{\phi_{dec}}$; trained CtRetroPromptPool $(\mathcal{K}, \mathcal{V}) = \{(\boldsymbol{k}_i, \boldsymbol{P}_i)\}_{i=1}^M$.

1: **procedure** TRAIN($\{s_{train}\}, (\boldsymbol{k}_i, \boldsymbol{P}_i)\}_{i=1}^M, \varphi, \alpha, C$)
2:    **for** epoch_id **in** total_epochs :
3:       Draw a mini batch $B$
4:       ▷ *For illustration purposes, we use batch size=1 here. The computation here can be easily extended to batch size $\geq 1$.*
5:       **for** $s_{[0,T]}$ **in** $B$ :
6:         Update the loss $\leftarrow$ CALCLOSS($s_{[0,T]}, f_{\phi_{enc}}, f_{\phi_{dec}}, \alpha$)
7:       Update $f_{\phi_{enc}}, f_{\phi_{dec}}$ by backpropagation.
8:       **if** epoch_id $\% C == 0$ : Update $(\boldsymbol{k}_i, \boldsymbol{P}_i)\}_{i=1}^M$ by backpropagation.
9: **procedure** CALCLOSS($s_{[0,T]}, f_{\phi_{enc}}, f_{\phi_{dec}}, \alpha$)
10:    $\mathcal{L} \leftarrow 0$.
11:    ▷ *Recursively compute the loss.*
12:    **for** $e@t$ **in** $s_{[0,T]}$ :
13:       ▷ *Compute the likelihood loss; the technical details can be found in Mei & Eisner (2017) and Yang et al. (2022).*
14:       $\mathcal{L}_{event} \leftarrow$ Take a sum of log intensity at the event time by calling CALCINTENSITY($s_{[0,t)}, e@t, f_{\phi_{enc}}, f_{\phi_{dec}}, (\boldsymbol{k}_i, \boldsymbol{P}_i)_{i=1}^M$).
15:       $\mathcal{L}_{non\_event} \leftarrow$ Integrate log CALCINTENSITY($s_{[0,t)}, e@t$) over inter-event time interval.
16:       $\mathcal{L}_{nll} \leftarrow \mathcal{L}_{non\_event} - \mathcal{L}_{event}$
17:       ▷ *Compute the matching loss.*
18:       $\mathcal{L}_{matching} \leftarrow \sum_{\mathrm{K}_{top-N}} \varphi(f_{\phi_{enc}}(e@t), \boldsymbol{k}_{r_j})$
19:       $\mathcal{L} \leftarrow \mathcal{L}_{nll} + \alpha\mathcal{L}_{matching}$
20:    **return** $\mathcal{L}$
21: **procedure** CALCINTENSITY($s_{[0,t)}, e@t, f_{\phi_{enc}}, f_{\phi_{dec}}, (\boldsymbol{k}_i, \boldsymbol{P}_i)_{i=1}^M$)
22:    $s_{[0,t]} \leftarrow$ Append $e@t$ to history $s_{[0,t)}$.
23:    Encode $s_{[0,t]}$ by $f_{\phi_{enc}}$ to generate the hidden state $\boldsymbol{h}_t$.
24:    Matching the index $r_j{}_{j=1}^N$ based on equation 7.
25:    Select Top-N prompts $\{\boldsymbol{P}_{r_i}\}_{i=1}^N$.
26:    Prepend $\{\boldsymbol{P}_{r_i}\}_{i=1}^N$ to $\boldsymbol{h}_t$ and pass to the decode $f_{\phi_{dec}}$ to generate the intensity $\lambda_e(t), e \in \{1, ..., E\}$.
27:    **return** $\lambda_e(t), e \in \{1, ..., E\}$
---

on a subset of $4800$ most active users with an average sequence length of $150$ and then end up with $K = 20$ event types.

**Amazon** (Ni, 2018). This dataset includes time-stamped user product review behavior from January 2008 to October 2018. Each user has a sequence of produce review events where each event containing the timestamp and category of the reviewed product, with each category corresponding to an event type. We work on a subset of 5200 most active users with an average sequence length of 70 event tokens and then end up with $K = 16$ event types.

**StackOverflow** (Leskovec & Krevl, 2014). This dataset has two years of user awards on a question-answering website: each user received a sequence of badges and there are $K = 22$ different kinds of badges in total. We work on a subset of xx most active users with an average sequence length of xx event tokens.

For the Taobao dataset, each task includes approximately 1 day of time;for the Amazon dataset, each task includes approximately 2 years of time; for the StackOverflow dataset, each task includes approximately 5 months of time. Table 1 shows statistics about each dataset mentioned above.

**Algorithm 2** PromptTPP at test time of the $\mathcal{T}$-th task.

---

**Input:** An event sequence $s_{[0,T]} = \{e_i @ t_i\}_{i=1}^I$. Trained base model with a encoder $f_{\phi_{enc}}$ and a decoder $f_{\phi_{dec}}$; trained CtRetroPromptPool $(\mathcal{K}, \mathcal{V}) = \{(\boldsymbol{k}_i, \boldsymbol{P}_i)\}_{i=1}^M$ and the score function $\varphi$.

**Output:** Sampled next event $\widehat{e}_{I+1} @ \widehat{t}_{I+1}$.

1: **procedure** DRAWNEXTEVENT($s_{[0,T]}, f_{\phi_{enc}}, f_{\phi_{dec}}$)
2:     $t_0 \leftarrow T; \mathcal{H} \leftarrow s_{[0,T]}$
3:     ▷ *Compute sampling intensity*
4:     $\{\lambda_e(t_j \mid \mathcal{H})\}_{j=1}^N \leftarrow$ SAMPLEINTENSITY($s_{[0,T]}, f_{\phi_{enc}}, f_{\phi_{dec}}, \{(\boldsymbol{k}_i, \boldsymbol{P}_i)\}_{i=1}^M$) for all $t_j \in (t_0, \infty)$
5:     ▷ *Compute the upper bound $\lambda^*$.*
6:     ▷ *Technical details can be found in Mei & Eisner (2017)*
7:     find upper bound $\lambda^* \geq \sum_{e=1}^E \lambda_e(t_j \mid \mathcal{H})$ for all $t_j \in (t_0, \infty)$
8:     **repeat**
9:       draw $\Delta \sim \text{Exp}(\lambda^*); t_0 \mathrel{+}= \Delta$           ▷ *time of next proposed event $\widehat{t}_{I+1}$*
10:      $u \sim \text{Unif}(0,1)$
11:     **until** $u\lambda^* \leq \sum_{e=1}^E \lambda_e(t_0 \mid \mathcal{H})$
12:    draw $\widehat{e}_{I+1} \in \{1, \ldots, E\}$ where probability of $e$ is $\propto \lambda_e(t_0 \mid \mathcal{H})$
13:    **return** $\widehat{e}_{I+1} @ \widehat{t}_{I+1}$
14: **procedure** SAMPLEINTENSITY($s_{[0,T]}, f_{\phi_{enc}}, f_{\phi_{dec}}, \{(\boldsymbol{k}_i, \boldsymbol{P}_i)\}_{i=1}^M$)
15:    Assume the last event in $s_{[0,T]}$ is $e @ t$
16:    Generate a list of sample times $\{t_j\}_{j=1}^N, t_j \geq T$.
17:    Compute the intensity at sample times $\lambda_e t_j \leftarrow$ CALCINTENSITY($s_{[0,t]}, e @ t, f_{\phi_{enc}}, f_{\phi_{dec}}, \{(\boldsymbol{k}_i, \boldsymbol{P}_i)\}_{i=1}^M$)
18:    **return** $\{\lambda_e(t_j \mid \mathcal{H})\}_{j=1}^N$

---

| DATASET | K | # EVT TOKENS TOTAL | AVG # EVT TOKENS PER SEQ | AVG # EVT TOKENS PER TASK | AVG # SEQ PER TASK |
|---|---|---|---|---|---|
| TAOBAO | 20 | 720,000 | 150 | 80,000 | 32 |
| AMAZON | 16 | 360,000 | 70 | 42,000 | 10 |
| STACKOVERFLOW | 22 | 240,000 | 60 | 43,000 | 12 |

Table 1: Statistics of each dataset.

## D.2 3S statistics and Experiment on Distribution Shift

We use the 3S (sum-of-squared-spacings) statistics proposed by Shchur et al. (2021) to depict the distribution of an event sequence in continuous time. Compared to the classical KS statistics (Lewis, 1965), it uniformly captures multiple properties of event sequence, such as total event count and distribution of inter-event times. Empirically, replacing the KS score with the 3S statistic consistently leads to a better separation between distributions generated by different TPPs. Please refer to the original paper (Shchur et al., 2021) for a detailed discussion.

For exploring the distribution shift in the Taobao dataset, we randomly sampled a thousand sequences of events and split them into 10 subsets by timestamps: each subset has approximately equal time horizon and is notated sequentially from 0-th to the 9-th subset. Then we follow the procedure in (Shchur et al., 2021) to compute the 3S statistics for each subset and illustrate the results in Figure 8a.

## D.3 Evaluation Setup

To set up the training and evaluation process, we partition Taobao and Amazon datasets into 10 consecutively rolling slides (namely 10 tasks) and partition the StackOverflow dataset into 6 rolling slides (namely 6 tasks). For the Taobao dataset, each slide covers approximately 1 day of time; for the Amazon dataset, each slide covers 2 years of time; for the StackOverflow dataset, each slide covers approximately 5 months time. The subset in each task is split into training, validation, and test sets with a 70%, 10%, 20% ratio by chronological order. Each task has no overlap in the test set. In such a setting, the total test set covers approximately 70% of data.

| MODEL | # PARAMETERS | | |
|---|---|---|---|
| | TAOBAO | AMAZON | SO |
| PRE-NHP | 23.3K | 23.4K | 23.3K |
| PRE-ANHP | 25.4K | 25.6K | 25.4K |
| RE-NHP | 23.3K | 23.4K | 23.3K |
| RE-ANHP | 25.4K | 25.6K | 25.4K |
| O-TPP | <1K | <1K | <1K |
| CL-NHP | 27.6K | 27.7K | 27.6K |
| CL-ANHP | 29.5K | 29.6K | 29.5K |
| PT-NHP | 26.2K | 26.3K | 26.2K |
| PT-ANHP | 27.8K | 27.0K | 27.8K |

Table 2: Total number of parameters for models trained on the three datasets.

We train and evaluate each task sequentially. Our evaluation setup is close to that used in real applications: train the model using a fixed length of historical data and evaluate the model using the following window of the data.

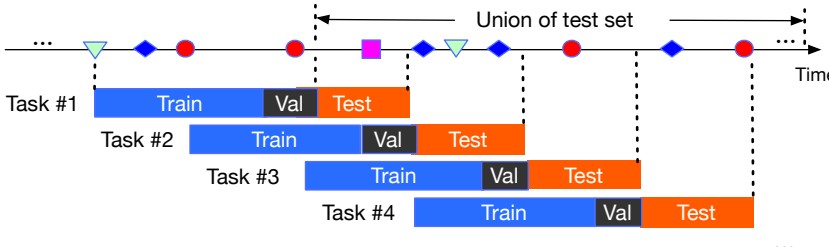

Figure 9: A demonstration of the sliding window validation method.

## D.4 Implementation Details

All models are implemented using the PyTorch framework (Paszke et al., 2017).

For the implementation of NHP, AttNHP, and thinning algorithm, we used the code from the public GitHub repository at https://github.com/yangalan123/anhp-andtt (Yang et al., 2022) with MIT License.

For O-TPP, as the authors Yang et al. (2017) have not published the code, we implement it using the tick (Bacry et al., 2017) library.

For CL-NHP and CL-ANHP, without the public code, by following the main idea of the authors Dubey et al. (2022), we develop the hypernetwork with an MLP layer and add apply regularizer to the hypernetwork parameters while learning a new event sequence, which prevents adaptation of the hypernetworks parameters completely to the new event sequence. Note that, the base models used are NHP and ANHP, respectively, instead of FullyRNN (Omi et al., 2019) applied in the original paper.

We implemented our methods with PyTorch (Paszke et al., 2017) and published the code at at https://github.com/yanyanSann/PromptTPP.

## D.5 Training and Testing Details

### D.5.1 Training and Hyperparameters Selection

**Training base TPP model.** To train the parameters for a given neural TPP, we performed early stopping based on log-likelihood on the held-out dev set.

- For NHP, the main hyperparameters to tune are the hidden dimension $D$ of the neural network. In practice, the optimal $D$ for a model was usually $32, 64, 128$, and we search for the optimal value among them for different datasets.

| MODEL | DESCRIPTION | VALUE USED | | |
|---|---|---|---|---|
| | | TAOBAO | AMAZON | SO |
| PRE-NHP | RNN HIDDEN SIZE | 76 | 76 | 76 |
| PRE-ANHP | TEMPORAL EMBEDDING | 64 | 32 | 64 |
| | HIDDEN SIZE | 64 | 64 | 64 |
| | LAYER NUMBER | 3 | 3 | 3 |
| RE-NHP | RNN HIDDEN SIZE | 76 | 76 | 76 |
| RE-ANHP | TEMPORAL EMBEDDING | 64 | 32 | 64 |
| | HIDDEN SIZE | 64 | 64 | 64 |
| | LAYER NUMBER | 3 | 3 | 3 |
| O-TPP | KERNEL SIZE | $20 \times 20$ | $16 \times 16$ | $22 \times 22$ |
| CL-NHP | RNN HIDDEN SIZE | 64 | 64 | 64 |
| CL-ANHP | TEMPORAL EMBEDDING | 64 | 32 | 64 |
| | HIDDEN SIZE | 64 | 64 | 64 |
| | LAYER NUMBER | 3 | 3 | 3 |
| PT-NHP | RNN HIDDEN SIZE | 64 | 64 | 64 |
| | $M$(RETRIEVAL PROMPT POOL SIZE) | 10 | 10 | 10 |
| | $N$(TOP-N SELECTED) | 4 | 4 | 4 |
| | $L_p$(PROMPT LENGTH) | 10 | 10 | 10 |
| | $C$(ASYNCHRONOUS REFRESH FREQUENCY) | 2 | 2 | 2 |
| PT-ANHP | TEMPORAL EMBEDDING | 64 | 32 | 64 |
| | HIDDEN SIZE | 64 | 64 | 64 |
| | LAYER NUMBER | 2 | 2 | 2 |
| | $M$(RETRIEVAL PROMPT POOL SIZE) | 10 | 10 | 10 |
| | $N$(TOP-N SELECTED) | 4 | 4 | 4 |
| | $L_p$(PROMPT LENGTH) | 10 | 10 | 10 |
| | $C$(ASYNCHRONOUS REFRESH FREQUENCY) | 2 | 2 | 2 |

Table 3: Descriptions and values of hyperparameters used for models trained on the three datasets.

- For AttNHP, in spite of $D$, another important hyperparameter to tune is the number of layers $L$ of the attention structure. In practice, the optimal $L$ was usually $1, 2, 3, 4$. In the experiment, we choose the hyperparameter based on the held-out dev set while keeping AttNHP to have a similar size to that of NHP.

**Training PromptTPP.** We find $\alpha$ in equation 11 is not sensitive and works well in a large range, so we set $\alpha = 0.1$ consistently for both datasets. For the prompts, we set $M = 10, N = 4, L_p = 10$ for both datasets. For the asynchronous training parameter $C$, we choose $C = 2$ for Taobao and Amazon datasets by default.

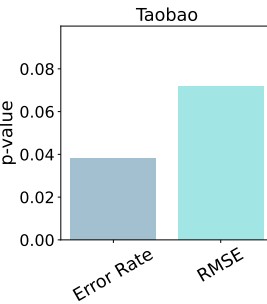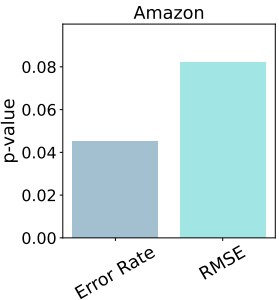

Figure 10: Statistical significance of the improvements from a temporal prompt on Taobao and Amazon datasets.

**Chosen Optimizer and Hyperparameters.** All models are optimized using Adam (Kingma & Ba, 2015). Table 3 contains descriptions that list all of the hyperparameters set throughout our experiments.

**Testing.** As described in Mei & Eisner (2017), we minimized the Bayes risk via the thinning algorithm to determine decisions for what a predicted next event time $\widehat{t}_{i+1}$ and type $\widehat{e}_{i+1}$ would be after conditioning on a portion of a sequence $s_{[0,t_i]} = [e_1@t_1, \ldots, e_i@t_i]$. All experimental results are averaged over 5 runs, and the corresponding standard deviation is reported as well. In the experiment, we report the metrics on the test set for each task as well as the average metrics over all the tasks.

**Integral Approximations.** During training and testing, there are a number of integrals (e.g., log-likelihood in equation 11) that need to be computed, which are not feasible in closed form. Thus, we must approximate them. All integrals and expectations are approximated via Monte Carlo (MC) estimates with certain amounts of samples used. $\mathcal{L}_{non\_event}$ in equation 1 uses 100 MC samples during training and testing. When evaluating the integrals used for next event predictions in thinning algorithm, we used 100 samples where the sample points were shared across integrals for a single set of predictions in order to save on computation. The exact approximation procedure for the log-likelihood can be found in Mei & Eisner (2017).

**Environment.** All the experiments were conducted on a server with 256G RAM, a 64 logical cores CPU (Intel(R) Xeon(R) Platinum 8163 CPU @ 2.50GHz) and one NVIDIA Tesla P100 GPU for acceleration.

### D.6  Analysis II Details: Statistical Significance

We performed the paired permutation test to validate the significance of our proposed temporal prompt. Particularly, for each model variant (Pt-NHP and Pt-ANHP), we split the test data into ten folds and collected the paired test results with temporal prompt and with the standard prompt, respectively, for each fold. Then we performed the test and computed the p-value following the recipe at https://axon.cs.byu.edu/Dan/478/assignments/permutation_test.php.

The results are in Figure 10. It turns out that, on both datasets, the performance differences are strongly significant for the error rate metric (p-value $< 0.05$) and weakly significant for the RMSE metric (p-value $\approx 0.08$).

### D.7  More Result: Computational Complexity of PromptTPP

The computational complexity comes from two parts: the TPP model and prompt pool.

Take ANHP as the base model. Assume the input sequence length is $K$, event embedding size $d_e$. Recall the prompt length $L_p$, selection size $N$ and prompt pool size $M$, key size $D$.

ANHP is attention-based, so its original complexity is $\mathcal{O}(K^2 d_e)$. By considering the attached prompt, the prompt-augmented ANHP's complexity becomes $\mathcal{O}((K + NL_p)^2 d_2)$. Retrieval's complexity is $\mathcal{O}(MD^2)$. So the total complexity is $\mathcal{O}((K + NL_p)^2 d_2 + MD^2)$.

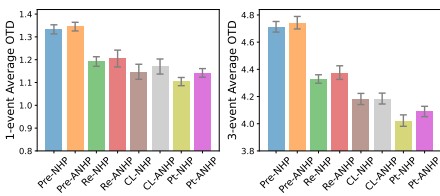
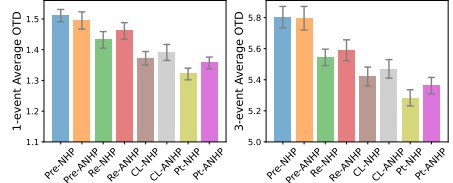

(a) Evaluation results on Task 9.

(b) Evaluation results on Task 10.

Figure 11: OTD distance comparison of generated sequence of all models on Amazon dataset.

## D.8 More Result: Generation Ability Comparison

We investigated the generative ability of the models empirically on Amazon dataset. Given a trained model, we fixed the prefix event sequence and performed the 1-event sampling and 3-event sampling (autoregressively) on the test set of task 9 and task 10. We followed (Xue et al., 2022) to compute the average optimal transport distance (OTD) to measure the distance between the generated sequence and the ground truth. Seen from Figure 11, our proposed models Pt-NHP and Pt-AttNHP achieves the best results. This is consistent with the findings in Main Results in the paper.

