# OpenReview forum: "Prompt-augmented Temporal Point Process for Streaming Event Sequence"
_NeurIPS.cc/2023/Conference — NeurIPS 2023 poster_

### Official Review · Reviewer_GihP · 2023-07-02

**Soundness:** 3 good
**Presentation:** 2 fair
**Contribution:** 3 good
**Rating:** 6
**Confidence:** 4

**Summary:**

The author introduces PromptTPP, a method that integrates classical neural temporal point processes with a continuous-time retrieval prompt pool. This approach enables the model to learn event streams sequentially, eliminating the need to buffer previous examples or task-specific features in the context of continual learning.

**Strengths:**

The paper's motivation for introducing this method is clear, and the experiments conducted to validate its effectiveness are relatively comprehensive.

**Weaknesses:**

The method description is not clear. The author needs to present the prompt-learning-based task in more detail. For example, the prompt length Lp, selection size N, and prompt pool size M are hyperparameters, but under common sense, the buffer of continual learning is dynamically changing. You can show the case study of the Taobao dataset and the Amazon dataset.

The combination of PPT, CL, and PL is somewhat unnatural. For example, I hardly see the idea of CL in the method.

Explain how the prompt learning method can help CL in terms of theoretical or technical details, rather than just experimental accuracy. Instead of a decline in catastrophic forgetting, it's possible that the decrease in average error is caused by an improvement in accuracy at each time point. Thus, the author also needs to show how the model performs on previous events after learning new events. If it is a part of the experiment, kindly explain it in detail.

Do the prompt settings need to be related to downstream tasks? If I understand correctly, your setting only concerns time, and it does not change depending on the process's intended goal. But in classical prompt learning, the prompt is customized, in NLP, adapting language model using downstream tasks. As you show in experiments, the task of each dataset is known beforehand, so will custom methods improve the results more than general methods? Can we investigate temporal prompt + event prompt?

**Questions:**

see above

**Limitations:**

see above

---

> ### Author Rebuttal · Authors · 2023-08-09
>
> > The method description is not clear.... the buffer of continual learning is dynamically changing. You can show the case study of the Taobao dataset and the Amazon dataset.
>
> In our experiment we found the performance w.r.t. hyperparameters is relatively stable over time. See below table. For each task,  increasing N from 1 to 2 brings ~0.8% improvement on error rate constantly; increasing N from 2 to 4 brings the change within [-0.02%, 0.06%] mostly.
>
> Amazon dataset, given $L_p=10, M=10$, sensitivity analysis of error rate w.r.t. selection size $N$ across all tasks
>
> |	|Task 0 	|Task 1	|Task 2	|Task 3	|Task 4	|Task 5|	Task 6	|Task 7	|Task 8 	|Task 9|	Avg|
> |-----|-----------|-----------|-----------|-----------|-----------|--------|-------------|-----------|-----------|-------|-------|
> |N=1	|75.04	|75.82|	73.29|	74.15|	68.51	|67.49|	67.83|	63.19|	63.65|	61.35|	69.03|
> N=2  	|74.41	|75.20	|72.79	|73.5|	67.89	|66.91|	67.30	 |62.81|	63.18|	60.779|	68.47|
> |improve rel%  | 0.847% |	0.824%|	0.687%|	0.884%|	0.913%|	0.867%|	0.788%	|0.605%	|0.744%	|0.939%	|0.811% |
>
> |	|Task 0 	|Task 1	|Task 2	|Task 3	|Task 4	|Task 5|	Task 6	|Task 7	|Task 8 	|Task 9|	Avg|
> |-----|-----------|-----------|-----------|-----------|-----------|--------|-------------|-----------|-----------|-------|-------|
> |N=2|	74.41	|75.20	|72.79	|73.50	|67.89	|66.91|	67.3	|62.81|	63.18	|60.779	|68.47|
> |N=4|	74.39	|75.16	|72.78	|73.45	|67.84	|66.87	|67.31	|62.77	|63.15|	60.74|	68.44|
> |improve rel %  |	0.027%|	0.053%	|0.014%	|0.068%	|0.074%	|0.060%|	-0.015%|	0.064%|	0.048%|	0.064%	| 0.045%|
>
> Therefore to better measure the impact of hyperparameters, we use the average performance as the metrics, same as in [1].  More details is in Analysis VI in section 4.2.
>
> [1] DualPrompt: Complementary Prompting for Rehearsal-free Continual Learning. ECCV 2022.
>
> > The combination of PPT, CL, and PL is somewhat unnatural. For example, I hardly see the idea of CL in the method.
>
> We consider the streaming setting for modeling event sequence. Through an empirical analysis, we validated that pretraining a model will fail to adapt to new data due to distribution shift in the data while retraining a model will suffer from catastrophic forgetting. See the analysis in Appendix A.
>
> So it is natural to approach the problem by adopting CL, an area studying how systems learn sequentially from a continuous stream of data. In the paper, we aim to build a CL model for such data in real scenarios with applicability and generality. Unfortunately, rehearsal-based methods are not applicable when there are resource constraints. Therefore we draw inspiration from PL, where we leverage prompts to sequentially learn knowledge and further store learned knowledge of event sequence in CL context.
>
> The idea of CL in our method mainly lies in our aim to make the model capable of continuously learning event sequences without catastrophic forgetting. This is done by integrating the TPP with a continuous-time retrieval prompt pool. The prompts are stored in a memory space and jointly optimized with the base TPP, ensuring the model learns event streams sequentially without buffering past examples or task-specific attributes.
>
> > Explain how the prompt learning method can help CL in terms of theoretical or technical details, ... Thus, the author also needs to show how the model performs on previous events after learning new events.
>
> Originated in NLP, PL reformulates learning downstream tasks from directly adapting model weights to designing prompts that “instruct” the model to perform tasks conditionally. A prompt encodes task-specific knowledge and has the ability to utilize pretrained models more effectively than ordinary finetuning [2].
>
> In our problem, we develop a module of temporal prompt that learns knowledge and further store the learned knowledge for event sequence in continuous time. We further structure the prompts in a key-value shared memory space called the retrieval prompt pool, and design a retrieval mechanism to dynamically lookup a subset of task-relevant prompts based on the instance-wise input of event sequences.  The optimized prompt pool ensures that shared (unselected) prompts encode shared knowledge for knowledge transfer, and unshared (selected) prompts encode task-specific knowledge that helps maintain model plasticity.
>
> [2] The power of scale for parameter-efficient prompt tuning. ACL 2021.
>
> > how the model performs on previous events after learning new events?
>
> Due to page limit,  we put it in [New results - how the model performs on previous events after learning new events] above in Author Rebuttal.
>
> > Do the prompt settings need to be related to downstream tasks? ... so will custom methods improve the results more than general methods? Can we investigate temporal prompt + event prompt?
>
> In most TPPs we do not consider downstream tasks. However, an extension of TPP is the recommender system when they consider continuous time information. In those cases, customized prompt may be appropriate to use because there are classical downstream tasks such as user identification etc. However, this is out of the scope of our paper.
>
> For the event prompt, we define it by
> - compute a histogram vector based on the normalized frequency of history event types.
> - add a MLP to map it and concatenate to form a prompt $P_{evt} \in \mathbb{R}^{L_p \times D}$ and concatenate $P_{evt}$ to one temporal prompt, and repeat the process to form a pool. Some resize layer is needed.
>
> However, on Amazon data, the improvement on most tasks is insignificant (p-value ~ 0.15) based on the permutation test. The possible reason is the learnable structure component in the temporal prompt, already learns to encode the information of events other than time, so adding an 'event prompt' has limited improvement on encoding the knowledge of the sequence. Given the shortness of the rebuttal period, we have not found a better custom method but we agree this is a direction that worths exploring.

---

> > ### Comment · Reviewer_GihP · 2023-08-18
> >
> > Thanks for the authors' response. I have updated my rating.

---

> > > ### Author Response · Authors · 2023-08-18
> > >
> > > Thank you very much for your support!
> > >
> > > We are thrilled to hear that our response has addressed your concerns. And we are committed to improving our presentation for the next version.

---

### Official Review · Reviewer_nDRf · 2023-07-04

**Soundness:** 3 good
**Presentation:** 3 good
**Contribution:** 3 good
**Rating:** 6
**Confidence:** 2

**Summary:**

This paper addresses the problem of Temporal Point Processes (TPP), which involves modeling continuous-time events. Specifically, the authors focus on developing a model for streaming data, where the model needs to be improved in a Continual Learning (CL) setting. To tackle this challenge, the authors propose a TPP model that combines the concepts of Prompt (which is modified for continuous time) and CL. The proposed method has demonstrated superior performance compared to alternative approaches on two datasets.

**Strengths:**

The paper is easy to read and is well-structured. The experiments conducted effectively demonstrate how each idea (Prompt and CL) contributes to the improvement over the alternative design.

**Weaknesses:**

I am not familiar with the related works on Neural TPP. However, I noticed that the experiment in this paper only utilizes two datasets, which might be considered somewhat limited. It is unclear if this is a common approach for papers working on this specific problem.

**Questions:**

1. Can the authors provide justification on using only two datasets in the experiment section?
2. In the abstract, the authors talk about privacy constraints associated with the TPP problem. Can authors provide more discussion on this with different TPP methods?


**Limitations:**

The paper includes a section discussing the limitations and potential societal impact.

---

> ### Author Rebuttal · Authors · 2023-08-09
>
> > Can the authors provide justification on using only two datasets in the experiment section?
>
> Firstly, using two benchmark datasets for a certain task is common in TPP papers. For example, [1] use MIMIC and StackOverflow to test their transformer structure on event sequences; [2] use Retweet and StockOverflow for interactive event modeling. our baseline [3] use Yelp/Meme for the zero shot event modeling.
>
> Secondly, a large number of datasets frequently used in TPP are not suitable for our problem. For example, we did not use Yelp/Meme because their average length is too short; Retweet is not suitable for our problem either because each sequence is not associated with an user ID or absolute timestamps so we can hardly make each sequence pertaining to a single individual. In our work, we processed two real datasets from the source,  which both have enough sequence lengths and clear user identifiers, therefore suitable for the CL problem in our context. We have performed extensive analysis on predictive performance, convergency, hyperparameters, ablation study etc.
>
> Lastly, also proposed by other reviewer, to further validate the effectiveness of our method, we perform the test on a new dataset. Please see [New results - new dataset StackOverflow] in Author Rebuttal above as the reply-to-all message. The results are consistent with our main results in the paper.
>
>
> [1] Transformer Embeddings of Irregularly Spaced Events and Their Participants, ICLR'22.
>
> [2] Bellman Meets Hawkes: Model-Based Reinforcement Learning via Temporal Point Processes, AAAI'23.
>
> [3] HyperHawkes: Hypernetwork based Neural Temporal Point Process, (updated version is accepted by KDD 2023).
>
>
> > In the abstract, the authors talk about privacy constraints associated with the TPP problem. Can authors provide more discussion on this with different TPP methods?
>
> Firstly, we clarify that the privacy constraints [4] mainly refers to common industrial cases where user data (detailed food/exercise/sleep/medical event log, their social media interactions, identification data) can not be stored in a buffer. In this context, rehearsal-based CL methods are not applicable because they rely on a rehearsal buffer to re-train a portion of past examples. Simply buffering past data and re-train the model does not work either.
>
> Our proposed method introduced a pool of continuous-time prompts to learn and store the learned knowledge for event sequences. We structure the prompts as a  retrieval prompt pool, and design a retrieval mechanism to dynamically lookup a subset of task-relevant prompts based on the instance-wise input of event sequences. The retrieval prompt pool, which is optimized jointly with the generative loss, ensures that shared (unselected) prompts encode shared knowledge for knowledge transfer, and unshared (selected) prompts encode task-specific knowledge that helps maintain model plasticity. This mechanism removes the necessity of a rehearsal buffer. Therefore our method is applicable in a more realistic CL setting, i.e., where privacy constraints exist.
>
> [4] Reza Shokri and Vitaly Shmatikov. Privacy-preserving deep learning. In Proc SIGSAC conference on computer and communications security, 2015

---

> > ### Comment · Reviewer_nDRf · 2023-08-16
> >
> > I am satisfied with the authors' response, and will update my original rating.

---

> > > ### Comment · Reviewer_nDRf · 2023-08-16
> > >
> > > Rating is updated to 6: Weak Accept

---

> > > > ### Author Response · Authors · 2023-08-18
> > > >
> > > > Thank you very much for your support!
> > > >
> > > > We are thrilled to know that our responses have positively affected your opinions.

---

### Official Review · Reviewer_26jC · 2023-07-07

**Soundness:** 2 fair
**Presentation:** 2 fair
**Contribution:** 2 fair
**Rating:** 6
**Confidence:** 3

**Summary:**

The paper proposes a novel prompt-augmented continual learning (CL) framework for modeling streaming event sequences, called PromptTPP. The framework incorporates a continuous-time retrieval prompt pool and a retrieval mechanism to dynamically lookup a subset of task-relevant prompts based on the instance-wise input of event sequences. The main contributions of the paper are (i) the introduction of PromptTPP, a novel prompt-augmented CL framework for neural TPPs, and (ii) the applicability and generality of the approach.

**Strengths:**

1. The use of a continuous-time retrieval prompt pool and a retrieval mechanism to dynamically look up a subset of task-relevant prompts based on the instance-wise input of event sequences is a novel approach to CL.
2. The paper is well-organized and easy to follow. The authors provide clear explanations of the proposed framework and the experiments. The figures and tables are also well-designed and easy to understand.
3. The PromptTPP addresses modelling streaming event sequences in a continual learning setting.

**Weaknesses:**

1. The paper does not provide a detailed analysis of the computational complexity of the proposed framework, which is an important consideration when working with large datasets and complex models.
2. The paper does not provide a detailed comparison with other state-of-the-art methods for modeling streaming event sequences in a continual learning setting, which would help researchers understand the strengths and weaknesses of PromptTPP in comparison to other approaches.

**Questions:**

1. Can the authors provide more details on the retrieval mechanism used in the proposed framework? Specifically, how does the retrieval mechanism select a subset of task-relevant prompts based on the instance-wise input of event sequences?
2.  How can the proposed approach be used for unwanted tracking of individual behaviour, and what steps can be taken to mitigate this risk?
3.  It would greatly enhance the understanding of the proposed framework if the authors could provide more in-depth information regarding the generative loss and its optimization. Specifically, how are the generative loss and retrieval prompt pool jointly optimized?

**Limitations:**

See the weaknesses above

---

> ### Author Rebuttal · Authors · 2023-08-09
>
> > The paper does not provide a detailed analysis of the computational complexity of the proposed framework.
>
> The computational complexity comes from two parts: the TPP model and prompt pool.
>
> Take ANHP as the base model. Assume the input sequence length is $K$, event embedding size $d_e$. Recall the prompt length $L_p$, selection size $N$ and prompt pool size $M$, key size $D$.
> - ANHP is attention-based, so its original complexity is $O(K^2 d_e)$. By considering the attached prompt, the prompt-augmented ANHP's complexity becomes $O((K+NL_p)^2 d_e)$.
> - Retrieval's complexity is $O(M D*D)$.
>
> So the total complexity is $O((K+NL_p)^2 d_e + M D^2)$.
>
> In the paper, we indeed provide a detailed analysis on the efficiency our method by looking at the convergency of learning curves (see Analysis IV in section 4.2), which is more common in TPP works [1,2,3]. The conclusion is that PromptTPP only adds a small number of parameters so that it generally has the same convergence rate as the base model. The asynchronous prompt optimization scheme improves the convergence (more remarkably on the Amazon dataset).
>
> [1] Neural Hawkes Process, NeurIPS 2017.
>
> [2] Transformer Hawkes Process, ICML 2020.
>
> [3] Noise-Contrastive Estimation for Multivariate Point Processes, NeurIPS 2020.
>
> > The paper does not provide a detailed comparison with other state-of-the-art methods for modeling streaming event sequences in a continual learning setting, which would help researchers understand the strengths and weaknesses of PromptTPP in comparison to other approaches.
>
> Firstly, in our experiments, we use two SOTA TPP as base models, one RNN-based and one attention-based, so that researchers could see how different base model affects the CL performance and choose appropriate base TPP for their own CL tasks.
>
> Secondly, so far the Dubey's work [4] is the only available neural TPP in CL setting. We introduce it in sec 4.1 and implement it as a baseline (notated as CL-NHP, CL-ANHP). In main results (section 4.2), we analyze the comparison between them and our method Pt-NHP/Pt-ANHP. Our methods outperform by a notable margin due to the novel design of CtRetroPromptPool.
>
> Compared to Dubey's work,
> - the strength of our method is, in the streaming setting, we have stronger ability to reduce catastrophic forgetting while adapting to new knowledge; our method has stronger applicability (the prompt pool and the event retrieval mechanism removes the necessity of a rehearsal buffer and knowing the task identity) and generality (our approach is general-purpose that it can be integrated with any TPPs);
> - the weakness is our method does not address zero shot event problem in CL setting, which his work focuses on (e.g., predict crimes for new cities while retaining and transferring information from old cities).
>
> [4] HyperHawkes: Hypernetwork based Neural Temporal Point Process, (updated version is accepted by KDD 2023).
>
> > Can the authors provide more details on the retrieval mechanism used in the proposed framework? Specifically, how does the retrieval mechanism select a subset of task-relevant prompts based on the instance-wise input of event sequences?
>
> How to select a subset of prompts based on the instance-wise input of event sequences? The input sequence decides which prompts to choose through query-key matching. Given a query $h_i$, the encoded event vector, we search for the closest keys over $K$ via MIPS.
>
> How to guarantee that a similar key value represents a similar event feature ？ See the loss function Equation 11, during the optimization, the second term pulls selected keys closer to corresponding query event in the retrieval process, so similar keys will represent similar events.
>
> How to validate the effectiveness of the key-value mechanism? See Analysis V in section 4.2. We remove the learnable key associated with prompts (w/o k-v in Figure 6b) and directly use the mean of prompts as keys. This strategy causes a moderate drop in performance.
>
> > How can the proposed approach be used for unwanted tracking of individual behavior, and what steps can be taken to mitigate this risk?
>
> Examples of event streams with potential social impact include a person’s detailed food/medical event log, their social media interactions, etc. Because our model is able to continually making more accurate predictions for user behavior, it could potentially be used for surveillance. This is a common societal impacts discussed in TPP [3,5]
>
> To mitigate this risk, rehearsal-free methods are preferred to use, as no meta data (user identification) of events needs to be stored in the buffer. This line of methods help prevent the leakage of private data. Other measures may include stricter privacy protection, which is out of the scope of our paper.
>
> [5] HYPRO: A Hybridly Normalized Probabilistic Model for Long-Horizon Prediction of Event Sequences, NeurIPS 2022.
>
> > It would greatly enhance the understanding of the proposed framework if the authors could provide more in-depth information regarding the generative loss and its optimization. Specifically, how are the generative loss and retrieval prompt pool jointly optimized?
>
> In equation 11, the loss functions has two terms: the first term is the typical generative loss of the event sequence and the second term refers to a surrogate loss. During optimization, the generative term fits the joint distribution of events while the surrogate term pulls selected keys closer to corresponding query event in the retrieval process, so similar keys will represent similar events. Besides, to accelerate training, we adopt the techniques in [6] to asynchronously update all embeddings in the prompt pool every $C$ training epochs. So in every epoch we update the parameters of TPP encoder/decoder by backpropagation; when epoch_id %C ==0, we update the prompt pool by backpropagation. See Algorithm 1 in appendix for full details.
>
> [6] Decoupling Knowledge from Memorization: Retrieval-augmented Prompt Learning, NeurIPS 2022.

---

> > ### Comment · Reviewer_26jC · 2023-08-20
> >
> > After carefully reviewing the authors' responses and the comments from other reviewers, I have decided to maintain my original rating.

---

### Official Review · Reviewer_n7nj · 2023-07-27

**Soundness:** 4 excellent
**Presentation:** 4 excellent
**Contribution:** 3 good
**Rating:** 7
**Confidence:** 2

**Summary:**

This paper aims to enhance the Neural Temporal Point Processes (TPPs) used in modeling continuous-time event sequences. Particularly in the real-world, the distribution of event data shifts over time, which existing TPPs fail to reflect. To address this, the authors propose PromptTPP, a continual learning framework that utilizes prompts. The effectiveness of this proposed method has been demonstrated through experiments with Taobao and Amazon datasets.

**Strengths:**

1. The paper is reader-friendly, with comprehensive equations and illustrative figures.
2. The proposed method is well-structured and intuitively compelling.
3. The experimental results convincingly demonstrate the effectiveness of the proposed approach.
4. The availability of the code simplifies the process of reproduction, making it easily accessible for further study.

**Weaknesses:**

While I'm not deeply familiar with this field, there don't seem to be significant weaknesses. However, I do have a concern related to the experiments conducted. The authors refer to CLTPP (Dubey et al., 2022) as the only neural TPP with continual learning (CL) capabilities. However, they did not conduct experiments on the Yelp and Meme datasets used in this paper (Dubey et al., 2022). Is there any reason for it?

**Questions:**

1. In line 112, it appears that the latter part of the sentence is missing. (`Compared with ordinary fine-tuning, literature shows […] In our`)
2. According to Figure 3, while the type error rate shows improvement (a decrease) for later tasks in the sequence, the time RMSE demonstrates the opposite trend, worsening (increasing) over time. What is the reason for this opposing trend between the two performance metrics?
3. I am interested in understanding the specific degree of distribution variances across tasks. The authors illustrate the KL divergence of each subsequent period relative to the 0-th period in Supplementary Figure 7. However, a more comprehensive comparison across all task pairs, possibly utilizing a confusion matrix, might yield more insightful and useful information.

---

> ### Author Rebuttal · Authors · 2023-08-09
>
> > The authors refer to CLTPP (Dubey et al., 2022) as the only neural TPP with continual learning (CL) capabilities. However, they did not conduct experiments on the Yelp and Meme datasets used in this paper (Dubey et al., 2022). Is there any reason for it?
>
> The problem formulation and dataset processing in our paper are different from Dubey's.
> 1. In our work, we focus on the streaming event setting: each sequence corresponds to a unique user's behavior over time. We truncate the dataset by time to make the tasks: each user's sequence and we study how a model can continuously learn to predict the behavior of the users.
> 2. In Dubey's work, they focus on zero-shot prediction in the CL setting, e,g., predict crimes for new cities while retaining and transferring information from old cities. So they do not necessarily require each sequence to be that of the same user.
>
> We don't use Yelp and Meme because they have too short average sequence length (both < 20)  user-wise. So we process two datasets by our own from the source.
>
> Lastly, to ensure the applicability of our proposed method, we perform the experiments on a newly processed dataset. Due to page limit, we put the results in [New results - new dataset StackOverflow] in Author Rebuttal above globally.  The findings are consistent with those on Taobao and Amazon datasets.
>
>
> > In line 112, it appears that the latter part of the sentence is missing.
>
> We want to say "Compared with ordinary fine-tuning, literature shows that prompt-augmented learning results in a sequence-based model having higher capacity to learn features [1,2,3,4]."
>
> Thanks for point this and we will fix it in the next version.
>
> [1] The Power of Scale for Parameter-Efficient Prompt Tuning，EMNLP 2021.
>
> [2] Pre-Train, Prompt, and Predict: A Systematic Survey of Prompting Methods in Natural Language Processing, ACM Computing Surveys 2022.
>
> [3] Learning to prompt for continual learning, CVPR 2022.
>
> [4] Prefix-tuning: Optimizing continuous prompts for generation, ACL 2021.
>
> >  According to Figure 3, while the type error rate shows improvement (a decrease) for later tasks in the sequence, the time RMSE demonstrates the opposite trend, worsening (increasing) over time. What is the reason for this opposing trend?
>
> In TPP, the event type follows categorical distribution while the time follows a continuous distribution. The distribution shift also evolves with time as the tasks are defined sequentially by time. We believe the event time prediction is more susceptible to the effects of distribution shift over time and the error may accumulate. As a result, the time RMSE worsens in later tasks.
>
> However, for time RMSE, at each task our method Pt-ANHP performs better than baselines with a notable margin, and this margin becomes larger in later tasks, showing the effectiveness of our method.
>
> To further validate our method on later tasks, we investigate the generative ability on the Taobao dataset. Given a trained model, we fixed the prefix event sequence and performed the 1-event sampling and 3-event sampling on the test set of task 8 and task 9. We followed [5] to compute the average OTD as the metrics to measure the distance between the generated sequence and the ground truth.  Taking ANHP as the base model, our proposed model Pt-ANHP achieves the best results.
>
> OTD of 1-event sampling (the lower the better)
> | Model | Task 8   |  Task 9  |
> |------|---------|----------|
> | Pre-ANHP     |   1.345      |  1.497        |
> | Re-ANHP     |   1.205      |    1.461     |
> | CL-ANHP     |   1.170      |    1.391      |
> | Pt-ANHP     |   1.142      |    1.357      |
>
>
> OTD of 3-event sampling (the lower the better)
> | Model | Task 8   |  Task 9  |
> |------|---------|----------|
> | Pre-ANHP     |   4.743     |  5.796        |
> | Re-ANHP     |   4.376      |    5.590     |
> | CL-ANHP     |   4.186      |    5.470      |
> | Pt-ANHP     |   4.090      |    5.362      |
>
>
> [5] HYPRO: a hybridly normalized probabilistic model for long-horizon prediction of event sequence, NeurIPS 2022.
>
> >  I am interested in understanding the specific degree of distribution variances across tasks. The authors illustrate the KL divergence of each subsequent period relative to the 0-th period in Supplementary Figure 7. However, a more comprehensive comparison across all task pairs, possibly utilizing a confusion matrix, might yield more insightful and useful information.
>
> We have computed the confusion matrix of Taobao below. Not surprising, KL increases with the time distance between the tasks, e.g., KL (1st and 9th) = 0.2208 > KL(1st and 4th)=0.1156. This further indicates the distribution shift over the time.
>
>
> |	  |Task 0 |	Task 1	|Task 2	|Task 3	|Task 4	|Task 5	|Task 6|	Task 7	|Task 8|	Task 9|
> |-------|--------|-----------|------------|-----------|------------|---------|---------|--------------|-------|----------|
> |Task 0	|0	|  0.1543	|0.1906	| 0.1873 |	 0.2228	|0 .2406	|0.2746	|0.2571	|0.2855|	0.2961|
> |Task 1	|	| 0	|0.0599	|0.1094	|0.1156	|0.1300	|0.1466|	0.1344|	0.1797|	0.2208|
> |Task 2	|	|	| 0 |	0.0202	|0.0619	|0.0728	|0.0901	|0.1198	|0.1311	|0.1722|
> |Task 3	|	|	|	| 0	| 0.0379	|0.0581	|0.0817|	0.0633|	0.1217	|0.1617|
> |Task 4	|   |	|	|	|	0	| 0.0314	|0.4166|	0.7303|	0.0993	|0.1138|
> |Task 5	|	|	|	|	|	|0	|0.0291	|0.0881|	0.6222	|0.1114|
> |Task 6	|	|	|	|	|	|	|0	|0.0363	|0.4210|	0.8474|
> |Task 7	|	|	|	|	|	|	|	|0	|0.0391	|0.4726|
> |Task 8	|	|	|	|	|	|	|	|	|0	| 0.0163|
> |Task 9	|	|	|	|	|	|	|	|	|	 |      0|

---

> > ### Comment · Reviewer_n7nj · 2023-08-18
> >
> > Authors have adequately addressed my concerns, so I will maintain my rating.

---

> > > ### Author Response · Authors · 2023-08-18
> > >
> > > Thanks very much for your response.
> > >
> > > We are thrilled to know that our response has addressed your concerns. We will improve the paper presentation based on all the reviews.

---

### Author Rebuttal · Authors · 2023-08-09

We thank all the reviewers for their constructive feedback. In this response, we will first use a few reply-to-all messages to clarify our dataset settings and present some new experiment results. Then we will provide a point-to-point response to each individual review. We can surely consolidate all the paper improvements (including clarified technical details, an improvement on writing, experiment results on new dataset, etc) in our final version.

###  Dataset settings

Reviewer n7nj and nDRf are concerned about the datasets we used. Here we clarify in a general way and also reply in more details in point-to-point responses, respectively.

Generally speaking, use two datasets for validating a certain model is common in TPP works[1,2,3]. In our works, we carefully preprocess two industrial and open sourced user behavior datasets and have done extensive analysis to support our conclusion.

We did not use the same datasets (Yelp and Meme) in Dubey's [4] work because they are not suitable for our problem: the sequence length is too short user-wise. In our problem, we aim to continuously learn the user behavior and formulate each sequence to be one unique user's timestamped behaviors and truncate by time into several tasks, so each sequence needs to be long enough. (Dubey's datasets do not have such requirement because he focuses on zero-shot event modeling in the CL setting, e.g., predict crimes for new cities while retaining and transferring information from old cities.  So their dataset is constructed category-wise rather than user-wise over time.

Lastly, we indeed process one more dataset and perform the experiment on it to show our method works across several real datasets. Please see [New results - new dataset StackOverflow].


[1] Transformer Embeddings of Irregularly Spaced Events and Their Participants, ICLR 2022.

[2] Bellman Meets Hawkes: Model-Based Reinforcement Learning via Temporal Point Processes, AAAI 2023.

[3] Neural Jump Stochastic Differential Equations, NeurIPS 2019.

[4] HyperHawkes: Hypernetwork based Neural Temporal Point Process, Dubey et al, (updated version is accepted by KDD 2023).

### New results - new dataset StackOverflow

Preprocess: StackOverflow [5] dataset has two years of user awards on a question answering website. Each user received a sequence of 22 kinds of badges in total. Following the processing recipe of previous work[1,6], we work on a subset of 4000 most active users with an average sequence length of 60 and then end up with K = 22 event types. We truncate the sequences by time into 6 tasks sequentially. Each task coverages approximately 5 months time .


Setup: we take ANHP as the base model, build Pt-ANHP, compare with three classical model Pre-ANHP, Re-ANHP, O-TPP and one CL-based model CL-ANHP.

Result: our method Pt-ANHP continuously outperforms others: the margin over the non-CL baselines (e.g., Re-ANHP) becomes larger over times (tasks) ; it is also outperforms CL-ANHP due to our design of CtRetroPromptPool which reduces catastrophic forgetting. The findings are consistent with those for Taobao and Amazon datasets.


error rate on each task (the lower the better)

| Model| Task 0| Task 1|Task 2|Task 3|Task 4|Task 5|Avg |
| ---- |---- |---- |---- |---- |---- |---- |---- |
| Pre-ANHP |50.99|  52.97	| 52.23| 	52.20| 	53.18| 	53.91| 	52.58|
|Re-ANHP| 	50.99| 	52.62	| 51.90	| 51.77	| 51.32| 	51.12| 	51.62|
|O-TPP	| 50.67	| 52.12	| 52.23	| 52.98	| 53.01	| 52.76| 	52.29|
|CL-ANHP| 50.76	| 51.93	| 51.23	| 51.16	| 50.36	| 50.27	| 50.95|
|Pt-ANHP| 50.72	| 51.30	| 50.73	| 50.08	| 49.44	| 49.01	| 50.21|


Time RMSE on each task (the lower the better)

| Model| Task 0| Task 1|Task 2|Task 3|Task 4|Task 5|Avg |
| ---- |---- |---- |---- |---- |---- |---- |---- |
|Pre-ANHP	|1.1109	| 1.1423	|1.1362	|1.1703	|1.1933	|1.2096	|1.1604|
|Re-ANHP|	1.1109	| 1.1365	|1.1321	|1.1455	|1.1562	|1.1690	|1.1417|
|O-TPP	| 1.1270	| 1.1461	|1.1456	|1.1653	|1.1772	|1.1969	|1.1596|
|CL-ANHP|	1.1201	| 1.1322	|1.1253	|1.1425	|1.1420	|1.1552	|1.1362|
|Pt-ANHP|	1.1204	| 1.1302	|1.1231	|1.1334	|1.1352	|1.1398	|1.1303|


[5] Leskovec, J. and Krevl, A. SNAP Datasets: Stanford large network dataset collection, 2014.

[6] HYPRO: a hybridly normalized probabilistic model for long-horizon prediction of event sequence, NeurIPS 2022.


### New results - how the model performs on previous events after learning new events

Request by reviewer GihP, we perform a test on Amazon dataset. We use ANHP trained on task 9 and Pt-ANHP continuously trained on task 9, evaluate them on previous tasks and see how the metrics changed. Specifically,
-  For ANHP,  evaluation on test set of task i (i<9):  ANHP trained on task 9 vs ANHP trained on task i.
-  For Pt-ANHP,  evaluation on test set of task i (i<9): Pt-ANHP trained until (including) task 9 vs Pt-ANHP trained until task i.

See from below tables, on both metrics, we see the drop in performance (error rate/rmse becomes higher) of Pt-ANHP is much less significant than ANHP: indicating our method stores well the knowledge of previous tasks,  which largely alleviates catastrophic forgetting.

Abs change on error rate:
|Model|	Task 0	|Task 1	|Task 2|	Task 3|	Task 4|	Task 5|	Task 6|	Task 7|	Task 8|
|-------| -------|-------|-------|-------|-------|-------|-------|-------|-------|
|ANHP|	+2.46	|+2.06|	+1.67|	+1.34|	+0.92|	+1.1|	+0.81|	+0.69|	+0.62|
|Pt-ANHP|	+0.63|	+0.5|	+0.64|	+0.42|	+0.28|	+0.48|	+0.2	|+0.07	|-0.04|

Abs change on time RMSE
|Model|	Task 0	|Task 1	|Task 2|	Task 3|	Task 4|	Task 5|	Task 6|	Task 7|	Task 8|
|-------| -------|-------|-------|-------|-------|-------|-------|-------|-------|
|ANHP	|+0.0213	|+0.0201	|+0.017|	+0.0094|	+0.0109|	+0.009|	+0.0061|	+0.0055|	+0.0031|
|Pt-ANHP | +0.0061	 |+0.006	 |+0.0053|	+0.005	|+0.0023	|+0.0035	|+0.0031	|+0.0012|	+0.0025|

---

### Decision · Program_Chairs · 2023-09-21

**Decision:**

Accept (poster)

**Comment:**

The paper aims to enhance the modeling of continuous-time event sequences using a novel framework called PromptTPP. The paper has received moderate-to-high ratings from the reviewers, with some concerns and questions regarding the method's clarity, computational complexity, dataset choices, and privacy considerations. The authors have addressed some of these concerns in their responses. Overall, the paper offers a valuable contribution to the field of neural temporal point processes.